# Exploration of schizophrenia-associated gene modules using graph theory, co-expression networks, and dimensionality reduction

Costas Bampos*, Vasileios Megalooikonomou

Computer Engineering and Informatics Department, School of Engineering, University of Patras, Patras, Greece

* costas.bampos@gmail.com

## Abstract

The PsychENCODE consortium has generated a comprehensive multi-omic dataset from human brain samples, spanning healthy individuals and those with neuropsychiatric disorders. In this study, we focus exclusively on schizophrenia and analyze PsychENCODE transcriptomic data to construct gene regulatory and co-expression networks, aiming to uncover biologically and clinically relevant gene modules. We apply three independent analytical approaches to the same dataset. First, we preprocess the data using Surrogate Variable Analysis (SVA) to correct for latent variation and normalize gene expression. For graph-based analysis, we use Pearson correlation and the igraph package to construct gene co-expression networks and apply Prim's algorithm to generate minimum spanning trees (MST) and compute centrality measures. Next, we implement Weighted Gene Co-expression Network Analysis (WGCNA), assuming a scale-free topology, to identify modules associated with schizophrenia traits. Dimensionality reduction is performed using Principal Component Analysis (PCA), with visualization aided by t-distributed Stochastic Neighbor Embedding (t-SNE) and Multidimensional Scaling (MDS). Functional enrichment is carried out using Gene Ontology (GO) and KEGG pathway databases. Each method reveals distinct yet complementary biological signatures associated with schizophrenia. The igraph-based approach highlights differentially expressed genes (DEGs) with high centrality, uncovering hub genes involved in mRNA export, synaptic signaling, and ion transport. WGCNA identifies co-expression modules strongly correlated with schizophrenia diagnosis, enriched for immune response, histone modification, and mRNA surveillance pathways. PCA isolates key genes contributing to diagnostic variance, with enrichment in neurotransmitter release cycles and cytokine signaling. Collectively, these results underscore the involvement of immune, synaptic, and epigenetic processes in schizophrenia and demonstrate the power of using multiple, orthogonal analytical lenses.

**Data availability statement:** All relevant data for this study are publicly available from the Synapse repository (https://doi.org/10.7303/syn4921369).

**Funding:** This study was supported by research funding from the project TAEDR-0539180 implemented within the framework of "Actions in interdisciplinary scientific areas with special interests for the connection with the productive fabric", Greece 2.0 - National Recovery and Resilience Plan to V.M.

**Competing interests:** The authors have declared that no competing interests exist.

## Introduction

### Overview

Schizophrenia is a chronic and debilitating neuropsychiatric disorder affecting approximately 1% of the global population. Characterized by cognitive dysfunction, delusions, and social withdrawal, its etiology is known to be multifactorial, involving complex genetic, epigenetic, and environmental interactions. Despite significant advances in neurogenomics, the molecular mechanisms underlying schizophrenia remain only partially understood. The PsychENCODE Consortium has assembled one of the most comprehensive multi-omic resources for the human brain, integrating transcriptomic, epigenomic, and genotypic data from both neurotypical individuals and patients diagnosed with major psychiatric disorders. Leveraging this rich dataset enables the identification of gene expression patterns and regulatory circuits potentially implicated in disease pathogenesis. However, the high dimensionality, batch effects [1], and biological heterogeneity inherent in brain tissue data present substantial analytical challenges. No single method can capture the full spectrum of molecular interactions relevant to complex diseases like schizophrenia. Therefore, in this study, we employ three independent and complementary computational strategies—graph-based network analysis (using the igraph R package) [2], Weighted Gene Co-expression Network Analysis (WGCNA) [3], and Principal Component Analysis (PCA) [4]—to interrogate gene expression data from PsychENCODE. Each approach offers a distinct perspective: igraph facilitates the identification of topologically central hub genes; WGCNA detects co-expression modules associated with clinical traits; and PCA highlights variance-driving genes across samples. By integrating insights across these methods, we aim to uncover disease-relevant gene networks and prioritize candidate genes for further investigation in the context of schizophrenia. In Fig 1 we present the integrated architecture of the three complementary computational methods we employed for disease module discovery.

The three analytic branches we mention complement one another: (i) the igraph/MST approach produces a sparse, visualizable backbone emphasizing topology and *node-level* prioritization (hubness, betweenness, potential control points); (ii) the WGCNA pipeline derives *module-level* co-expression structure, eigengenes, and module–trait relationships that are robust to noisy edges; and (iii) PCA (with t-SNE/MDS visualizations) provides a *variance-driven* view that identifies the genes that contribute most to sample separation and therefore to population-level phenotypes.

### Data acquisition and preprocessing

The PsychENCODE Consortium (PEC) has created a comprehensive online resource by processing human brain samples from both healthy controls and individuals affected by various neuropsychiatric disorders. This dataset, integrated with reprocessed data from other major sources such as ENCODE, CommonMind, GTEx, and the Epigenomics Roadmap, includes full genotyping; bulk and single-cell RNA sequencing (RNA-seq); chromatin immunoprecipitation sequencing (ChIP-seq); assay for transposase-accessible chromatin using sequencing (ATAC-seq); Hi-C; expression quantitative trait loci (eQTL)

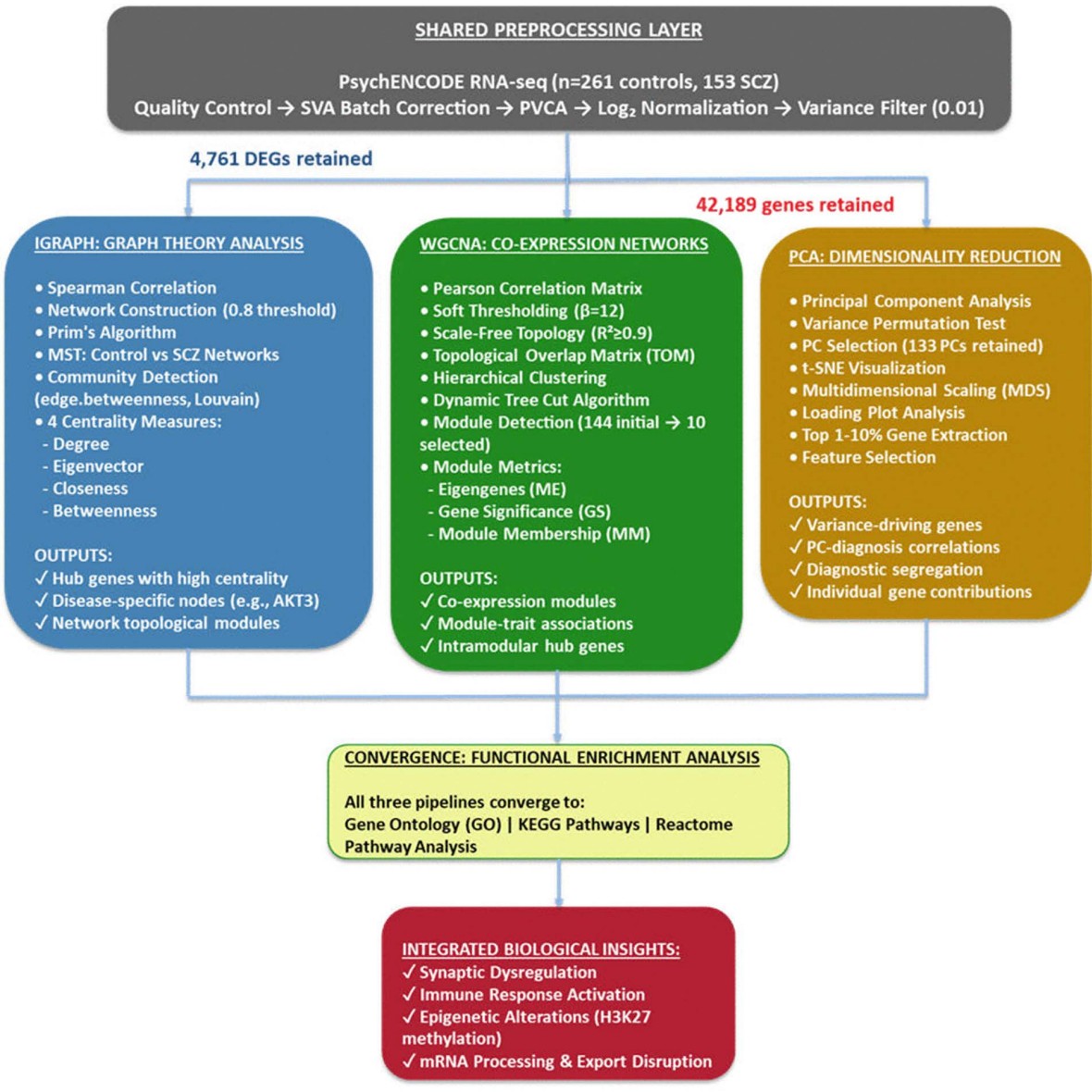

**Fig 1. Integrated three-pipeline analysis.** Flowchart showing from top to bottom: inputs accessed from PsychENCODE, preprocessing steps (SVA [5], PVCA [6], outlier removal), the three analytic branches deployed (igraph/MST centrality, WGCNA modules, PCA/t-SNE) and the criteria used for multi-axis target ranking. Finally, at the bottom of the flowchart we present the insights of the analysis. Sample sizes: controls n = 261, schizophrenia n = 153.

and chromatin QTL (cQTL); active enhancers; differentially expressed genes and transcripts; novel non-coding RNAs; patient metadata and phenotypes; as well as fully processed epigenomic signals and peak calls. The PsychENCODE resources and integrative analyses that made these harmonized datasets publicly available are described in [7,8].

## Analytical framework

In the present study, we implement three distinct techniques on PEC data to construct gene regulatory networks (GRNs), each leveraging different aspects of prior biological information:

- We construct graphs using only differentially expressed genes (DEGs) as vertices and transcriptional gene-to-gene (TG–TG) linkages as edges, then compute various centrality measures. For this task, we utilize the igraph R package.

- We analyze all genes without incorporating prior biological knowledge, except for the assumption of scale-free connectivity. This approach is used to detect correlation patterns, identify gene modules, summarize them by calculating eigengenes or identifying intramodular hub genes, examine relationships between modules and disease, and compute module membership metrics. We implement this approach using the Weighted Gene Co-expression Network Analysis (WGCNA) method R software package.

- In gene profiling studies, the majority of genes often represent noise that must be filtered out. In this study, we use PCA in a versatile manner—both as a preprocessing step and as a primary analytical tool—to identify the most significant genes contributing to each principal component and to detect their involvement in schizophrenia (SCZ)-related communities.

### Novelty and contribution

We acknowledge that immune, synaptic, and epigenetic dysregulation in schizophrenia are well-documented pathogenic pathways. Our contribution is not to re-discover these broad pathways or to claim a single novel causal gene; rather, it lies in three complementary innovations that together increase confidence in candidate mechanisms and therapeutic priorities.

1. Orthogonal integration of three independent computational methods revealing non-redundant biological signatures supported by the literature: The key innovation lies not in applying individual methods, but in demonstrating how three orthogonal approaches reveal non-redundant, biologically complementary signatures of schizophrenia—each capturing distinct aspects of disease pathology that would be missed by any single method alone. In contrast to the highly integrated model presented by Wang et al., [7] our approach intentionally disentangles these techniques to explicitly explain which method contributes to which detected aspect of the disease. This strategy significantly increases methodological explainability, an advantage that is not immediately obvious in fully integrated frameworks.

2. Discovery and transparent handling of previously undocumented rRNA-depletion protocol heterogeneity in PsychENCODE: Explicit identification, characterization, and correction of previously unrecognized technical confounding in PsychENCODE data—namely, heterogeneous rRNA-depletion protocols that cascade into artifactual batch effects masquerading as biological signal.

3. Multi-dimensional ranking system combining network topology, co-expression membership, phenotypic correlation, and druggability: we evaluate each therapeutic target gene across four independent criteria, rather than relying on just one. Their importance and influence within the disease-specific gene regulatory network topology, their membership and relevance within disease-associated co-expression gene modules, their direct correlation with clinical phenotypes such as diagnosis status, and their potential druggability—meaning they possess biological features or existing pharmacological evidence making them feasible targets for therapeutic intervention. This is a systematic, rigorous approach to therapeutic target prioritization which represents good practice in precision medicine.

These elements collectively define our methodological novelty: a reproducible, network-aware pipeline that couples conservative preprocessing and technical-confound correction with orthogonal analytics and multi-axis target ranking to prioritize robust, biologically plausible therapeutic hypotheses.

## Methods

### High-level overview

- **igraph / MST (node-level topology)** — Figs 3–12: MST visualizations, centrality distributions, community detection results, and node-level hub prioritization (see Methods: "Designing Gene Networks using igraph").

- **PCA / t-SNE / MDS (variance-driven)** — Figs 13–16: principal component analyses, PC–phenotype correlations, PC-based gene lists used for enrichment, and t-SNE/MDS visualizations (see Methods: "PCA").

- **WGCNA (module-level co-expression)** — Figs 17–22: soft-threshold diagnostics, module detection, module–trait correlations, GS vs MM scatterplots, and module-level enrichment (see Methods: "WGCNA: construction and parameters").

- **SVA / PVCA (preprocessing diagnostics)** — S2 File, S1–S7 Figs: outlier detection, surrogate variable selection, PVCA variance decomposition and protocol-heterogeneity diagnostics, pickSoftThreshold and permutation diagnostics.

## Initial preprocessing

We used the PsychENCODE Capstone processed data for all analyses. Specifically, the analyses relied on the following Capstone files: `DER-02_PEC_Gene_expression_matrix_TPM.txt`, `DER-01_PEC_Gene_expression_matrix_normalized.txt`, `Controls_only_PEC_Gene_expression_matrix_normalized.txt`, and the sample-level mapping file `PEC_capstone_data_map_clinical.csv`. The mapping file contains the sample metadata and phenotype fields (diagnosis, brain region, RIN, library protocol and other QC flags) that were used to derive the analytic cohort reported in this manuscript. Exact file versions and Synapse accession IDs for the Capstone release are available from the PsychENCODE resource (http://resource.psychencode.org/); that portal also documents how to obtain the limited, publicly accessible demographic/clinical fields and the process for requesting access to any restricted phenotype variables under PsychENCODE's data-use policies.

The submitted `PEC_capstone_data_map_clinical.csv` mapping file indicates that the PsychENCODE Capstone release contains control and pathogenic labelled samples in the clinical map (diagnosis field), along with per-sample identifiers (e.g., Synapse individualID, SynapseRNAseq_specimenID, specimenID) and basic covariates (sex, ethnicity, ageDeath, study). Our analysis began from the PsychENCODE "capstone" metadata CSV, which in the raw file contains 986 Control and 558 Schizophrenia entries. We first loaded and subset this table to retain sample identifiers and phenotype variables. We then restricted the dataset to samples present in the expression matrix, excluded all non-SCZ diagnostic categories (e.g., Bipolar, Autism, Affective, BP), removed rows with missing metadata, and finally applied an outlier/sample-quality mask derived from hierarchical clustering.

These successive filtering steps — intersection with the expression matrix, exclusion of other diagnoses, removal of incomplete metadata, and outlier exclusion — reduced the raw 986/558 counts to the final sets used in downstream analyses and tree visualizations: 261 control samples and 153 schizophrenia samples.

## SVA/PVCA diagnostics and SV selection

To identify and correct latent technical variation, we combined Principal Variance Component Analysis (PVCA) and Surrogate Variable Analysis (SVA). The preprocessing pipeline consisted of the following steps:

- We computed PCA on the full expression matrix and correlated the leading principal components with known covariates.

- We ran PVCA to quantify the variance explained by measured factors, including sex, age, brain region, RIN, library protocol, and sequencing center.

- We estimated surrogate variables (SVs) using SVA with model pairs, defining the full model as covariates plus diagnosis, and the null model as covariates only (S2 File, S1 Fig).

- We retained SVs only if generalized linear models confirmed they were not significantly associated with our biological variable of interest, diagnosis (S2 File, S4 Fig).

- We then re-ran PVCA, including the retained SVs, to confirm the successful reduction in residual variance (S2 File, S5 Fig).

 

During this procedure, we identified protocol heterogeneity—specifically, the use of different rRNA-depletion kits—that materially affected a subset of PCs (S2 File, S6 Fig). Where kit metadata were available, we included the depletion method as an explicit covariate; otherwise, the effect was captured by the retained SVs. All model matrices, SV selection logic, and diagnostic plots are provided in the deposited R scripts.

## WGCNA construction and parameters

In our analysis, we used WGCNA to construct signed gene co-expression networks and identify modules of co-expressed genes. Network construction used covariate-corrected expression values, with Pearson correlation utilized to estimate pairwise relationships. A critical parameter in this process is the soft-thresholding power, which influences the scale-free topology of the resulting network. Following previously established guidelines [3], we evaluated candidate powers and selected the soft-thresholding power at the point where the scale-free fit index first reaches 0.9.

Notably, our initial analysis indicated that the scale-free topology fit index did not exceed 0.8 prior to covariate correction. These findings suggested the presence of a strong underlying driver that rendered a subset of samples globally distinct—an observation supported by the pre-correction PCA plots (S2 File, S2 Fig shows PCA before/after covariate correction). This global heterogeneity, driven by the previously discussed technical artifacts (rRNA depletion protocols), resulted in high spurious correlations that violated scale-free assumptions. Following SVA/PVCA correction, the scale-free topology fit index successfully surpassed the 0.9 threshold and mean connectivity dropped appropriately, indicating successful mitigation of the confounding signal (diagnostic plots showing the scale-free fit index and mean connectivity are provided in S2 File, S7 Fig).

Once the adjacency matrix was computed, it was converted to a topological overlap matrix (TOM). Hierarchical clustering was performed on TOM dissimilarity, and modules were identified with a dynamic tree cut followed by merging using a module eigengene distance threshold. Module membership (MM) and gene significance (GS) were computed in standard fashion; hub genes were prioritized by combining MM, GS, and node-level centrality metrics where applicable.

The exact preprocessing, parameter choices, and diagnostic outputs used for all WGCNA-based analyses are consolidated below:

1. **Input matrix:** Variance-thresholded expression matrix after SVA-based covariate correction (quasi-constant genes removed with variance cutoff = 0.01, reducing the number of genes from 57,820–42,189).

2. **Outlier handling:** Sample outliers were removed following hierarchical clustering and the dendrogram cut rule described in S2 File, S3 Fig.

3. **Surrogate variables:** SVs were estimated using SVA with full and null model pairs; only SVs not associated with diagnosis were retained (SVA selection diagnostics: S2 File, S4 Fig).

4. **Soft-thresholding:** pickSoftThreshold guided selection; final soft-power $\beta$ = 12 (scale free fit $R^2 > 0.9$). Soft-threshold diagnostic plots are in S2 File, S7 Fig.

5. **blockwiseModules parameters (exact):** softPower = 12, minModuleSize = 50, deepSplit = 4, mergeCutHeight = 0.1, maxBlockSize = 16000.

6. **Module filtering and hub selection:** modules with consistent eigengene–trait correlations were retained; intramodular hub genes defined by $MM > 0.80$ and $GS > 0.20$ (see Methods and Supplementary module tables in deposited archive).

7. **Diagnostics and reproducibility:** All WGCNA R scripts, pickSoftThreshold outputs, module membership tables, and the full list of modules (n = 144 pre-filter, n = 10 prioritized) are included in the deposited code archive and Supplementary Files (see Supporting Information).

## Designing gene networks using igraph

**Preprocessing.** Gene co-expression network analysis groups together genes with similar expression profiles into modules of highly correlated, or co-expressed, genes. These gene modules are typically assumed to represent functionally related units [9], and even genes with poorly characterized roles may have their functions inferred through a guilt-by-association (GBA) approach—based on the premise that genes sharing expression patterns are likely involved in similar biological processes [10].

To construct and analyze a gene co-expression network, it is essential to evaluate the interactions between genes. These interactions are typically assessed by estimating the correlation between different gene expression profiles. Two commonly used methods for calculating correlation coefficients are Pearson correlation and Spearman's rank correlation.

The Pearson correlation coefficient is the most widely used measure of pairwise similarity in co-expression studies and serves as the default in popular network analysis tools. However, Pearson correlation has several limitations:

1. It assumes that the underlying data are normally distributed;

2. It captures only linear relationships between variables; and

3. It is highly sensitive to outliers, which can distort correlation estimates [11].

In contrast, the Spearman correlation coefficient, which is based on ranked values, is better suited for non-normally distributed data and is more robust to outliers [12].

In our study, the gene expression data were log-transformed in an effort to approximate a normal distribution. To verify the normality of the data, we examine normal quantile-quantile (Q-Q) plots of the gene expression values. However, even after log transformation, many genes do not follow a normal distribution. (see "Igraph: Constructing gene co-expression networks" in Results).

**Constructing gene co-expression networks.** To construct the DEG (differentially expressed gene) graph, we follow a series of steps:

First, we adjust the gene expression data for batch effects using SVA. Next, we scale the covariate-corrected data to avoid clustering highly expressed genes together, regardless of their actual expression patterns across samples. Preprocessing details (variance thresholding and DEG selection) are described in Methods ("WGCNA: construction and parameters"); here we summarize downstream network structure.

The data are then log2-transformed, and we compute the correlation matrix using Spearman correlation coefficients. We use the correlation matrix to generate the adjacency matrix. To enhance network specificity and account for the inherent asymmetry of RNA-seq distributions, we apply an empirical, distribution-aware threshold. Specifically, we remove edges with correlation weights that fall between 80% of the maximum positive correlation and 80% of the minimum negative correlation, preserving only the extreme regulatory tails. We further simplify the graph by eliminating multiple edges between nodes. Additionally, we discard single unconnected nodes and small disjoint components, retaining only the largest connected component.

In the context of constructing gene co-expression networks, Prim's algorithm (using the igraph R package [2]) was selected over Kruskal's algorithm due to its superior performance on dense graphs and its adaptability to incremental construction from a selected starting node—a feature particularly useful in biological network modeling. While both Prim's and Kruskal's algorithms are classical methods for finding a MST, they differ in approach: Kruskal's algorithm sorts all edges and builds the MST by selecting the smallest edge that does not form a cycle, making it efficient for sparse graphs. In contrast, Prim's algorithm grows the MST from a chosen node by continuously adding the nearest neighboring vertex, which is advantageous in dense networks—a typical property of gene co-expression matrices where thousands of genes may be highly interconnected. Furthermore, Prim's algorithm can be more memory-efficient in this context, as it does not require sorting all edges upfront and can be optimized using priority queues. These properties make Prim's algorithm

more suitable for large-scale, biologically realistic graphs where the number of potential gene–gene correlations is high and where preserving local structure during expansion is biologically meaningful [13].

Crucially, by extracting the MST using the raw, signed correlation weights, Prim's algorithm inherently prioritizes the strongest negative correlations. This process maps the transcriptomic repressor network, isolating mutually exclusive regulatory states and inhibitory pathways that standard absolute-weight algorithms often marginalize. By retaining only these essential repressive connections, the MST approach reduces noise while preserving a rigid topological structure that identifies functionally relevant bottleneck and hub genes. This is particularly important in high-dimensional gene expression data where spurious correlations can obscure biologically meaningful relationships. The alternative—retaining all edges above a correlation threshold—would create a dense, highly interconnected network that violates scale-free assumptions and obscures identification of true regulatory hubs.

**Prim's algorithm and spanning tree construction.** The analysis pipeline begins with preprocessing of gene expression data, including regression of surrogate variables to mitigate unmeasured confounding and removal of quasi-constant genes using a fixed variance cutoff. Pairwise gene–gene similarity is then quantified using Spearman rank correlation, producing a dense, weighted, undirected similarity matrix. At this stage, multiple viable options for network construction are available in principle, including retaining the full weighted graph, applying a hard correlation threshold to obtain a sparse but cyclic network, or adopting a soft-thresholding strategy (as implemented in WGCNA). For the final network representation used in downstream visualization and centrality analysis, we selected a MST constructed from the filtered Spearman correlation graph using absolute edge weights. This choice was motivated by the practical and interpretive goals of the study rather than by the assumption that MST captures the full biological complexity of the system. The MST enforces a unique, cycle-free backbone that retains exactly one strongest path connecting every pair of genes, yielding a compact and fully connected structure that can be visualized and interrogated at genome scale. Compared with denser weighted graphs, this representation makes dominant pairwise relationships, bridge genes, and interpretable paths immediately accessible without introducing additional, arbitrary pruning steps.

We generate two separate trees: one for 261 control samples and one for 153 schizophrenia samples. For layout visualization, we use the Large Graph Layout (layout_with_lgl) (Figs 3 and 6). For community detection we evaluate two algorithms: edge.betweenness.community and cluster_louvain. To emphasize gene proximity within the network, we further refine the layout using the Fruchterman–Reingold force-directed algorithm (layout_with_fr), combining contract and simplify functions from igraph.

**Centrality measures.** Centrality measures are widely used to characterize the importance of individual vertices within a network and have found broad application across diverse fields, including epidemiology, power grid analysis, gene co-expression networks, traffic control, and social network analysis. In the context of unweighted networks, several centrality metrics—such as degree, betweenness, closeness, and eigenvector centrality—have been extensively developed and applied. Each of these measures offers distinct insights into network topology and can complement one another in analysis.

For instance, degree centrality captures the immediate connectivity of a node and is useful for identifying locally influential vertices. In contrast, closeness centrality reflects the average shortest path from a node to all others, providing a more global measure of accessibility. Betweenness centrality quantifies the extent to which a node lies on the shortest paths between other nodes, and it is particularly useful for detecting potential bottlenecks or control points in the network—especially since it remains applicable even in disconnected graphs. Eigenvector centrality, on the other hand, evaluates a node's influence based on the centrality of its neighbors, offering insight into the broader hierarchical structure of the network.

We compute several centrality measures: degree, eigenvector, closeness, and betweenness. To facilitate comparison, we scale the centrality scores across all modules in both the control and schizophrenia networks and re-draw the network trees. Each module is represented by its top-ranking gene, with the gene's label size and node diameter scaled

proportionally to its centrality score. As a result, only the most central—and thus most influential—genes within the network are prominently visible.

## PCA

In gene expression profiling studies, only a small subset of genes (covariates) is typically associated with the response variables, while the majority represent noise that should be filtered out. Principal Component Analysis (PCA), one of the earliest and most widely used dimensionality reduction techniques in bioinformatics [4], is commonly employed to address this issue. PCA operates on the assumption that the variation in the original data can be effectively summarized using a limited number of metagenes—principal components (PCs)—which are linear combinations of all genes. These PCs are orthogonal to one another and represent a much lower-dimensional space than the original set of variables.

PCA is particularly effective at capturing both biological and technical sources of variability, which can then be compared against known covariates.

**Plotting.** To determine the optimal number of principal components (PCs) to retain for downstream analysis using t-SNE, we first compared the variance explained by each observed PC to that explained by PCs derived from a permuted version of the gene expression matrix. This approach allowed us to identify components that captured meaningful biological signal rather than noise. To validate our selection, we used the retained PCs as features in a random forest classifier to assess their predictive utility.

Subsequently, we applied t-SNE for dimensionality reduction and visualization, using the TSNE class from the scikit-learn library. In parallel, we performed multidimensional scaling (MDS) in R using the cmdscale function and compared the resulting visualization to the t-SNE output.

**Gene enrichment through PCA.** PCA plots were evaluated both before and after covariate correction, with individual data points color-coded according to known biological phenotypes, including age, ethnicity, sex, brain region of origin (e.g., hippocampus, dorsolateral prefrontal cortex), and diagnosis. To investigate the contributions of specific genes to the observed variance, we generated loading plots displaying the relationship between the top genes and their corresponding principal components.

To identify the most influential genes driving variation across the top PCs, we utilized the plotLoadings function from the PCAtools R package [14], applying the rangeRetain parameter to extract genes falling within the top and bottom 1% and 10% of loadings for the first five PCs.

In addition, we used the biplot functionality in PCAtools to visualize PC1 versus PC2 and computed Pearson's $R^2$ to assess the strength of association between each of the top PCs and clinical diagnosis. This step was performed after parameterizing the number of surrogate variables (SVs) included in the model, adjusting the proportion of gene expression variance explained, and deciding whether to z-transform the data.

Finally, genes with the highest contributions to the top five PCs were subjected to pathway enrichment analysis using the Reactome database to identify biological processes associated with schizophrenia.

## Enrichment analysis

The final step across all three methodological pipelines involved the functional annotation of the resulting gene sets. Enrichment analysis remains a widely adopted strategy for this purpose [15], with the Gene Ontology (GO), KEGG, and Reactome databases among the most commonly used resources. We employed the clusterProfiler R package [16] to perform overrepresentation analysis for GO terms and KEGG pathways, as well as enrichment analysis using the ReactomePA package [17], which implements the enrichPathway() function to map selected genes to Reactome pathways. All enrichment procedures utilized a hypergeometric test to evaluate whether the observed number of genes associated with a given category was significantly higher than expected by chance.

For annotation, only clusters with at least 10 genes were considered. Significance thresholds were set at a Benjamini–Hochberg adjusted $p-value < 0.05$ and a $q-value < 0.1$ to control for false discovery. To facilitate comparative analysis of functional profiles across gene clusters, we used the compareCluster() function in clusterProfiler, which computes enrichment results for multiple clusters and compiles them into a unified object for downstream analysis.

For visualization, we retained only the top 10 most significant functional categories per cluster. Enrichment results were presented using bar plots, dot plots, and enrichment maps, the latter of which represent functional categories as network nodes connected by edges indicating shared genes. In dot plots, dot color represents the adjusted p-value (with red indicating stronger enrichment and blue indicating weaker enrichment), while dot size corresponds to the number of genes in the enriched category. These visualizations help highlight functionally relevant processes and facilitate interpretation of the biological roles of the gene clusters.

## Results

### Igraph

**Constructing gene co-expression networks.** To evaluate the distribution of gene expression data, we constructed normal quantile-quantile (QQ) plots, comparing each gene's expression profile to that of a standard normal distribution (Fig 2). In these plots, normally distributed data align along a straight diagonal line. However, the observed data deviated from this line, following a nonlinear curve—indicative of a skewed, non-normal distribution typical of RNA-seq data.

Figs 3–5 and Figs 6–8 illustrate the sequential steps undertaken to construct the control and pathogenic gene co-expression networks, respectively. The objective of this process was to derive well-defined gene modules, each assigned a unique identifier to facilitate downstream functional enrichment analysis. For Gene Ontology (GO) enrichment, we systematically evaluated a range of parameters to characterize different aspects of the resulting networks. Specifically, we tested the effects of applying various community detection algorithms, weighted versus unweighted graph representations, and different gene selection criteria within each module. More specifically, for clustering we used two community detection algorithms on undirected, not weighted graphs: Edge Betweenness and Louvain-Leiden. Based on centrality scores, from each community detected, all genes were included for further analysis; or the top 10/30% of genes were included based on closeness and/or eigenvector and/or degree and/or betweenness (for results see Table 1).

Functional enrichment analysis of Gene Ontology (GO) terms indicated that the most prominent biological themes in the pathogenic networks were associated with synaptic function and immune-related processes (Table 2).

**Centrality measures distribution.** In this section, we constructed graph and spanning tree representations of differentially expressed genes (DEGs) in schizophrenia (SCZ) patients and their corresponding genes in control samples. Following community structure analysis for both tree objects, we estimated node-level statistics, including centrality measures, and conducted comparative analyses. We observed that centrality in the control network was more evenly distributed across nodes, whereas the pathogenic network exhibited a more concentrated distribution, with a few nodes assuming highly prominent roles.

A key consideration for downstream analysis was identifying genes that are central in the pathogenic network but of secondary importance in the control network. Such genes represent potential therapeutic targets, as their modulation could affect disease-related processes with minimal risk of off-target effects in healthy individuals. For instance, comparison of Figs 9 and 10 highlight ENSG00000117020 (AKT3) as a central node in the pathogenic circuit, particularly within the largest community (Community 2 in Fig 8), which is enriched for mRNA export from the nucleus. Interestingly, although AKT3 is highly connected and functionally influential, it is not formally included within this module, suggesting it may serve as a regulatory hub affecting the community without being a core component. This distinction presents a unique opportunity for indirect modulation of the disease module.

However, AKT3 also appears as a relatively central node in the control network (Fig 9), suggesting potential side effects if targeted indiscriminately. These findings underscore the importance of context-specific network analysis. Moreover, the significance of Community 2, despite being the largest in the pathogenic network, was marginal ($p \approx 0.04$), raising

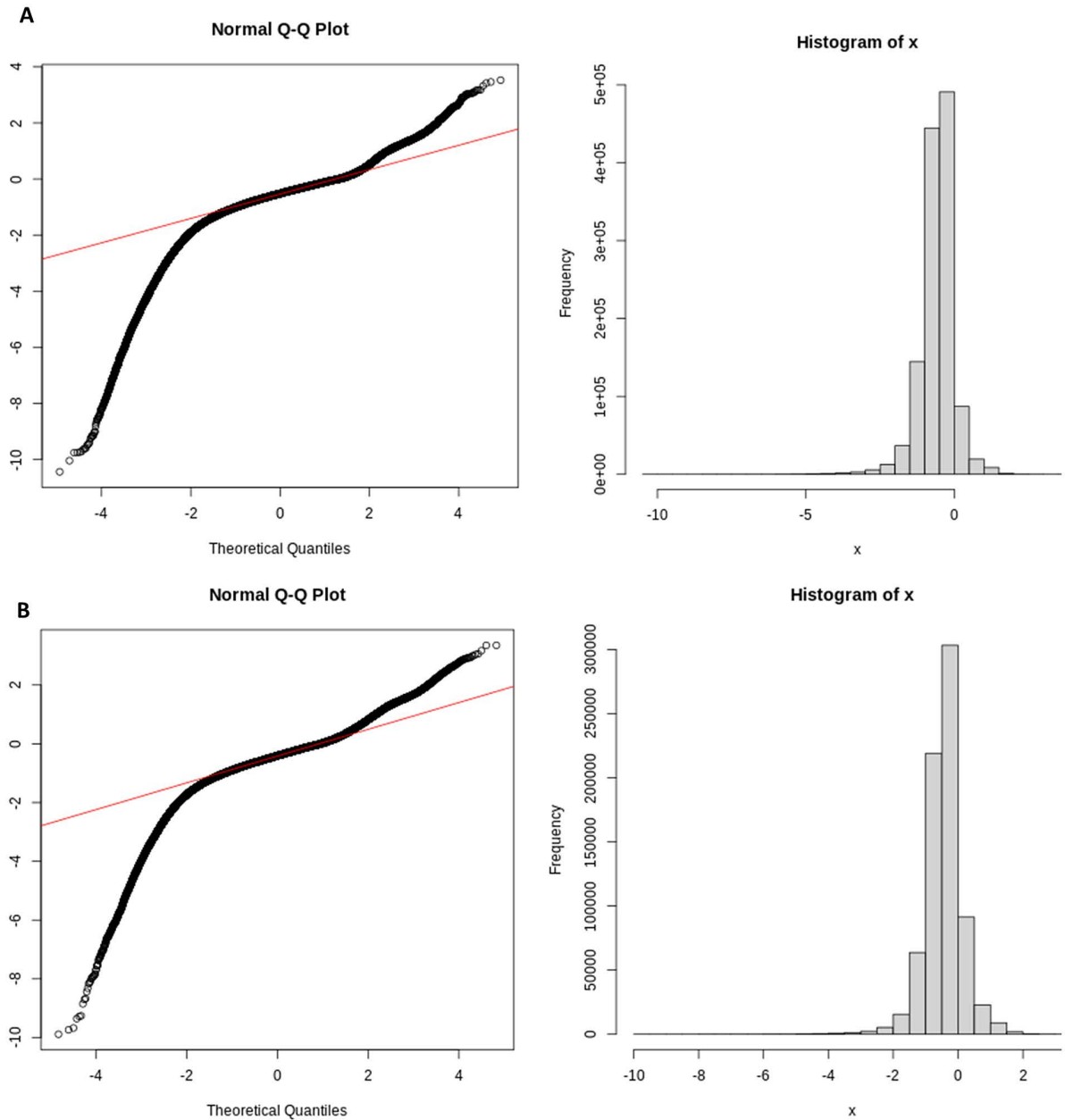

**Fig 2. Normal probability (Q–Q) plots and histograms for gene expression distributions in control (A) and schizophrenia (B) samples.**

concerns about the robustness and reproducibility of this module. Therefore, interpretations must be made with caution, and further validation is warranted to confirm its biological relevance.

In Figs 9 and 10, we normalized the centrality scores across all modules within the control and pathogenic gene co-expression networks, respectively to ensure comparability across metrics. Each module was then represented by the gene exhibiting the highest centrality score within that specific module. Both the label size and node diameter of each representative gene were proportionally scaled to reflect its relative centrality value.

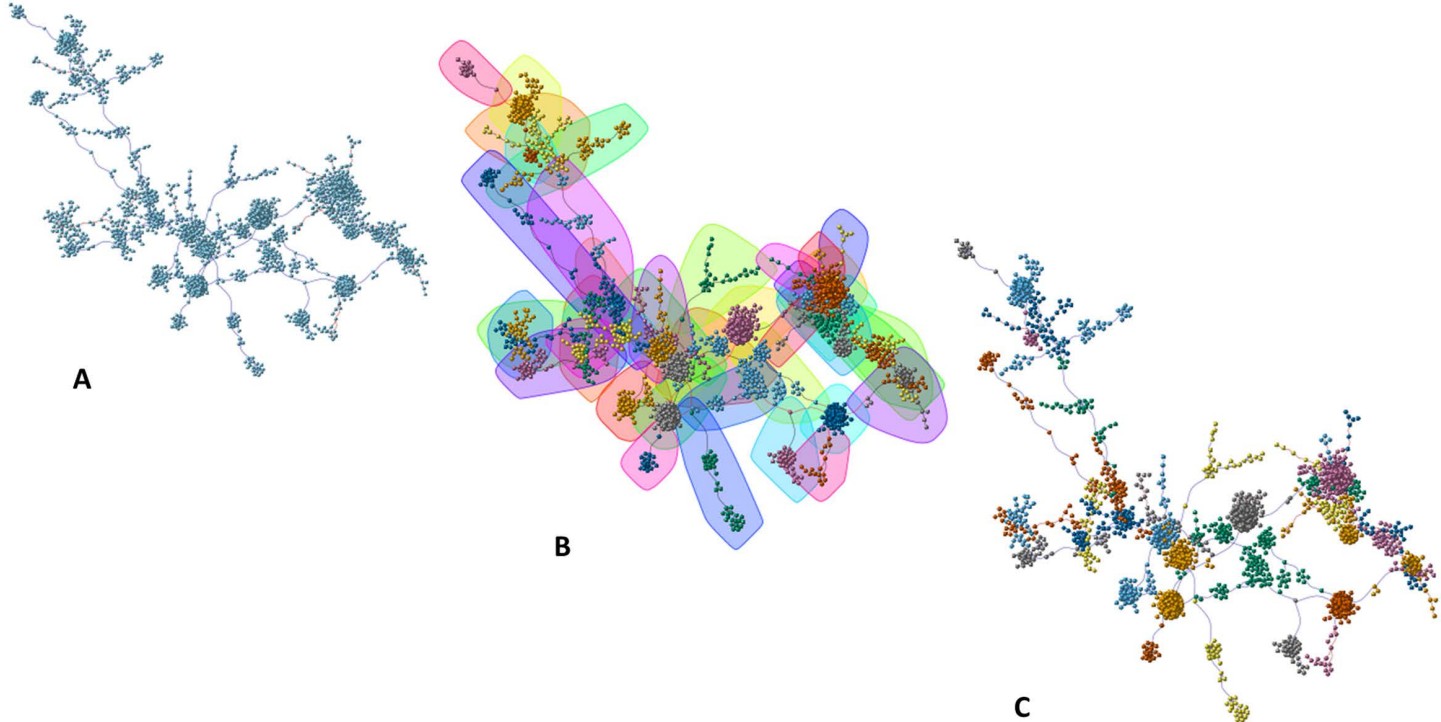

**Fig 3. Control network: module detection workflow. (A)** MST derived from Spearman correlations (see Methods for exact edge-pruning thresh-olds and component filtering). **(B)** Community detection (edge-betweenness and Louvain results shown). **(C)** Module color assignment used for downstream enrichment. Node colors indicate module membership. See Methods for layout algorithms and thresholds (Large Graph Layout and Fruchterman–Reingold).

Centrality measures are widely used to characterize the importance of individual vertices within a network and have found broad application across diverse fields, including epidemiology, power grid analysis, gene co-expression networks, traffic control, and social network analysis. In the context of unweighted networks, several centrality metrics—such as degree, betweenness, closeness, and eigenvector centrality—have been extensively developed and applied. Each of these measures offers distinct insights into network topology and can complement one another in analysis.

For instance, degree centrality captures the immediate connectivity of a node and is useful for identifying locally influential vertices. In contrast, closeness centrality reflects the average shortest path from a node to all others, providing a more global measure of accessibility. Betweenness centrality quantifies the extent to which a node lies on the shortest paths between other nodes, and it is particularly useful for detecting potential bottlenecks or control points in the network—especially since it remains applicable even in disconnected graphs. Eigenvector centrality, on the other hand, evaluates a node's influence based on the centrality of its neighbors, offering insight into the broader hierarchical structure of the network.

In Figs 11 and 12, we present histograms of these centrality measures for both the pathogenic and control networks, allowing a direct comparison of vertex importance distributions across the two biological conditions.

## Principal component analysis (PCA)

**Gene enrichment through PCA.** In Fig 13, we show plots of individual samples color-coded by group, accompanied by concentration ellipses to highlight potential gene-driven sample segregation based on diagnosis,

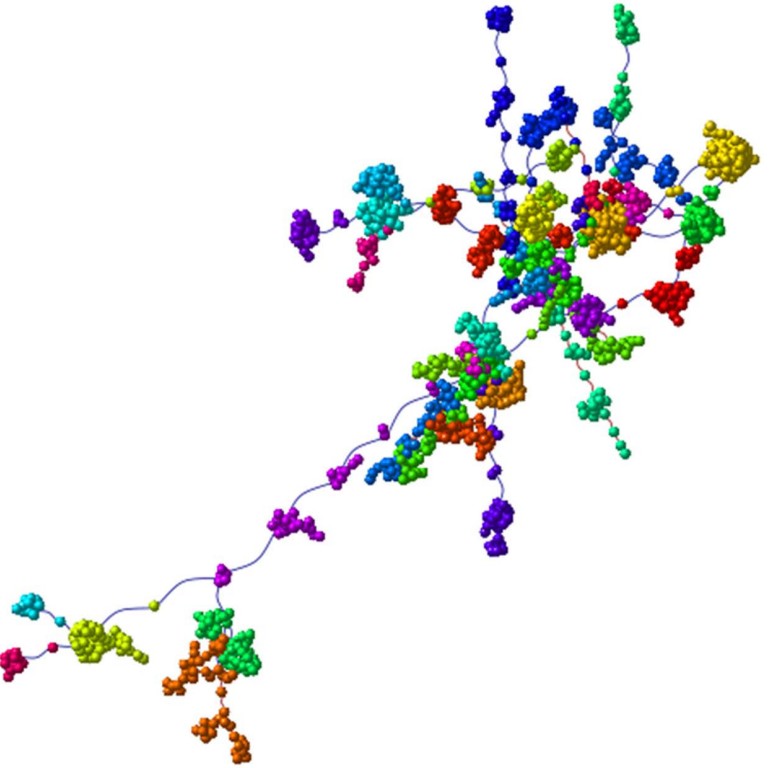

**Fig 4. Control network: module visualization.** Each module is represented by a different color.

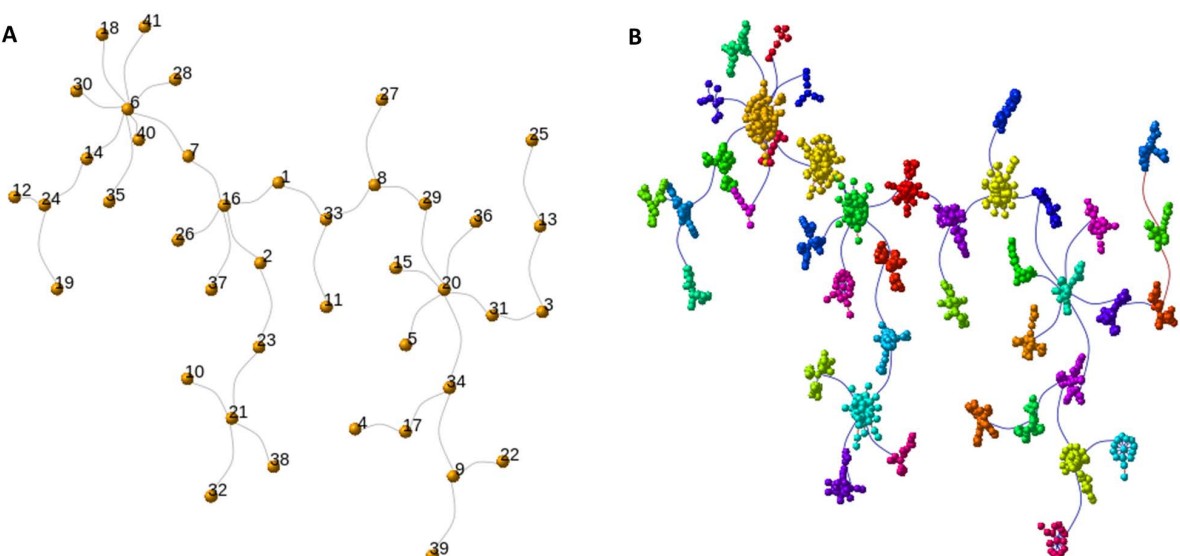

**Fig 5. Control network: layout refinement and community numbering.** (A) community numbering annotated on representative nodes; (B) scaled layout for visualization using Fruchterman–Reingold.

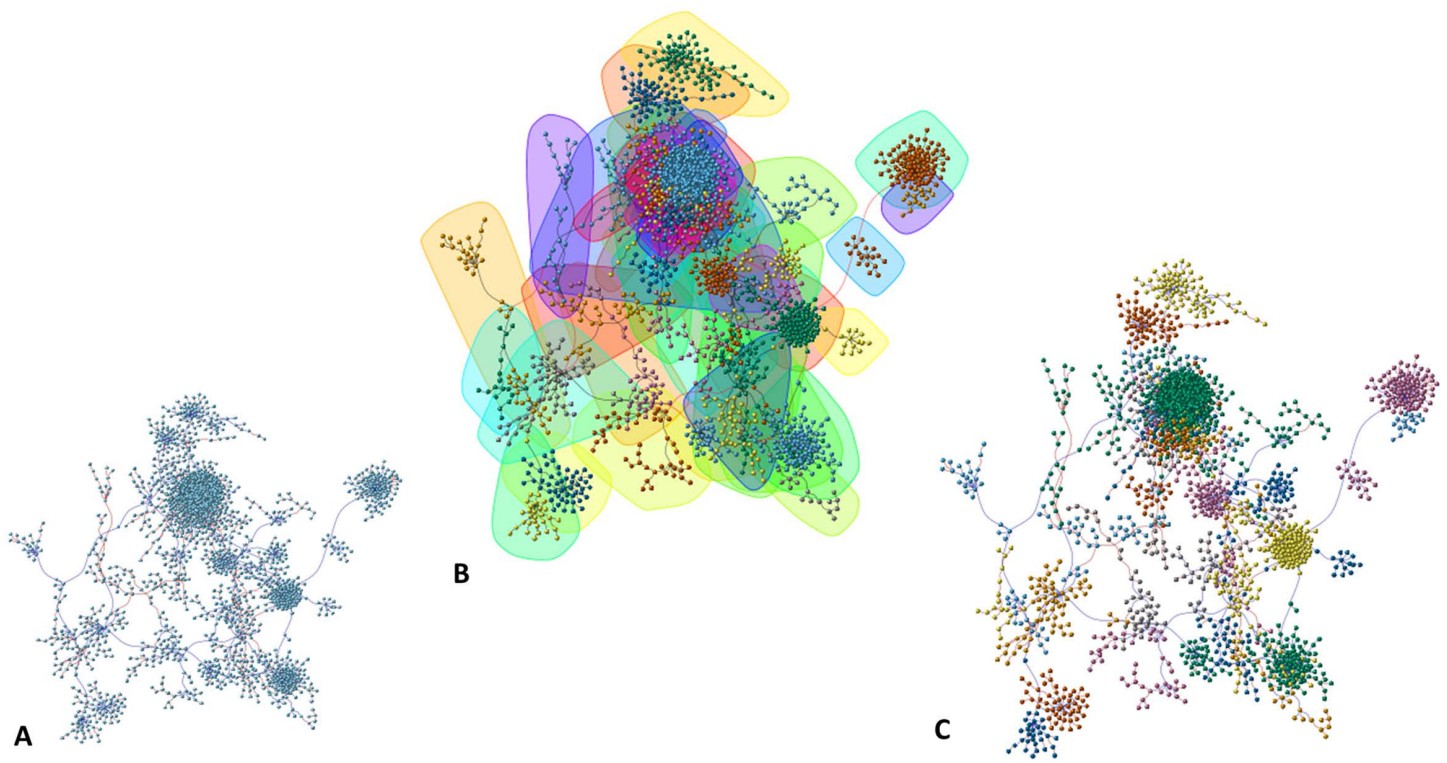

**Fig 6. Schizophrenia (pathogenic) network analysis.** (A) input network (undirected, null weights); (B) gene clustering results; (C) community detection via edge-betweenness.

age, sex, ethnicity, or brain region. The PCA plots reveal that among the examined biological phenotypes—sex, ethnicity, diagnosis, brain region, and age—age has the most pronounced influence on the overall data structure. This is evident from the clearer separation of age groups in the PCA plot. In contrast, other factors such as sex, ethnicity, diagnosis, and brain region show substantial overlap in PCA space, indicating that they contribute less to the major sources of variance captured by the first two principal components. Overall, these findings suggest that age is a significant driver of variation in the dataset, while the effects of the other phenotypes are more subtle or not well captured by the principal components.

Fig 14 illustrates the top 1% (A) and top 10% (B) of outlier genes associated with the top five PCs. These top-ranked genes were subsequently used for enrichment analysis.

Although PCA plots are primarily used to assess the effectiveness of covariate correction, we extended our analysis by examining the correlations between the top six principal components (PCs)—which together explain approximately 14% of the total variance—and various phenotypic traits. This approach aimed to identify which PCs were significantly associated with key variables, particularly diagnosis. As illustrated in Fig 15, PC1 exhibited the strongest correlation with diagnosis, making it the most notable component in this context. Additionally, PC2 showed a noteworthy association with lower age, particularly among individuals with a mean age of 20 years.

Gene set enrichment analysis revealed that the genes contributing most significantly to the variance captured by PC1 and PC2 belonged to two biologically informative groups. These gene sets were associated with synaptic function and inflammatory processes, respectively (Fig 15). To gain deeper insight, we repeated the analysis using

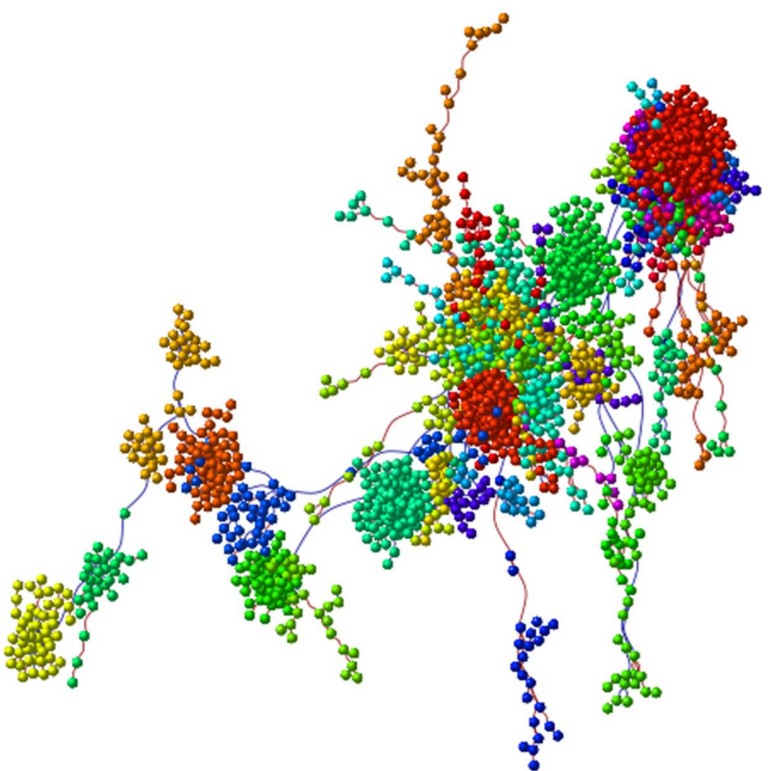

**Fig 7. Network analysis in schizophrenia: Different modules are represented by different colors.**

alternative parameters and performed Reactome pathway enrichment analysis (Table 3). This revealed several pathways of interest, including the P2Y/purinergic receptor signaling pathway, previously implicated in schizophrenia (SCZ) [18] (Fig 1a in S1 File). Additional enriched pathways included: regulation of FOXO transcription factor localization via AKT-mediated phosphorylation; Chk1/Chk2 (Cds1)-mediated inactivation of the Cyclin B:Cdk1 complex; activation and mitochondrial translocation of BAD, which is linked to immune response and apoptosis; and the RAP1 signaling pathway (Fig 1b in S1 File).

**tSNE.** t-SNE can offer certain advantages over traditional PCA when analyzing biological data. Prior to applying t-SNE, we first reduced the dimensionality of the original dataset to suppress noise and improve the accuracy and efficiency of pairwise distance estimation between samples. Fig 16A and 16B illustrate the process of selecting the number of principal components (PCs) to retain as input for t-SNE. PCs whose variances fall within the mean of the permuted distribution (indicated by the red line) are considered to represent random noise and were excluded. In contrast, the minimal set of informative PCs lies above the intersection point of the observed variance (green line) and the permuted variance (red line).

Based on this criterion, we determined that 133 PCs should be retained for subsequent t-SNE visualization. To validate this selection, we performed random forest classification using only the 133 retained PCs, as shown in Fig 16C. The final t-SNE plot, based on these components, is presented in Fig 16D.

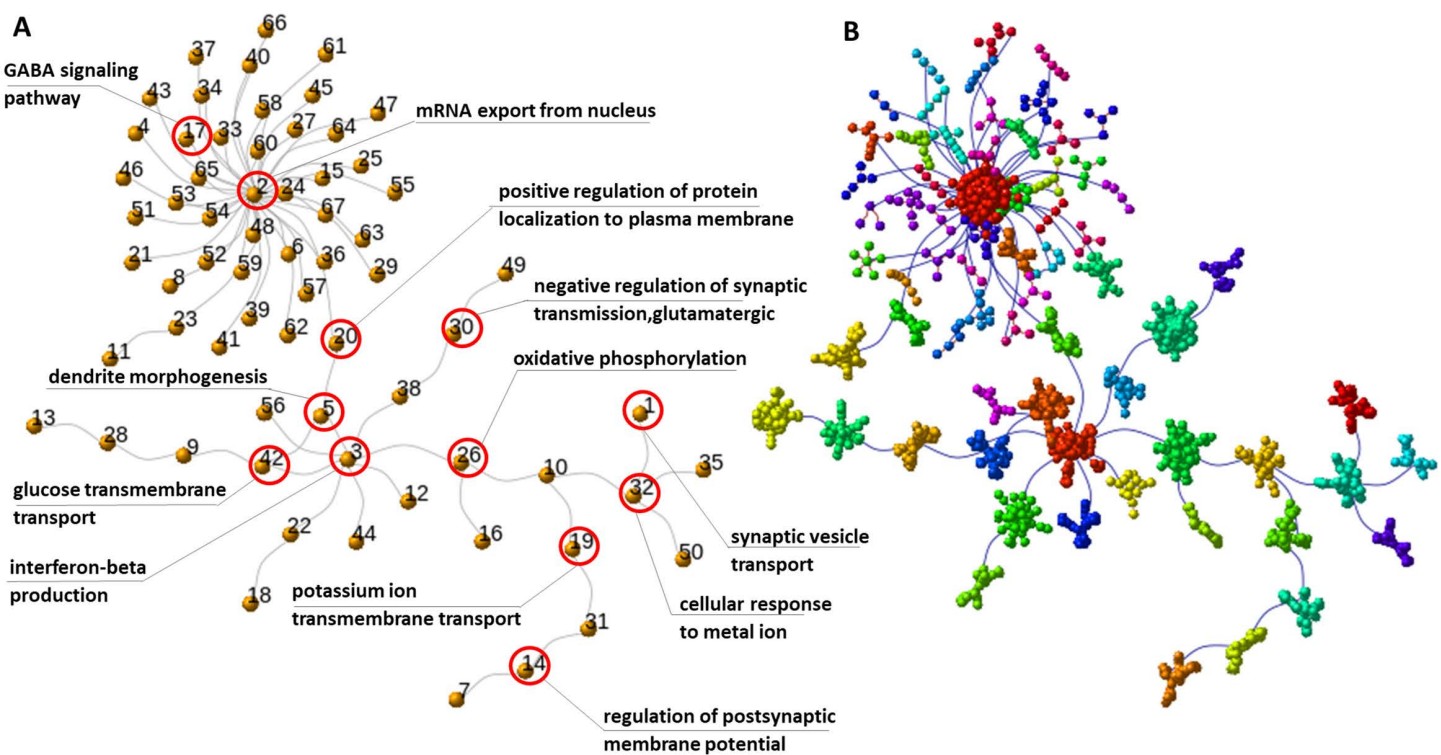

**Fig 8. Network analysis in schizophrenia: We scaled the network for better visualization (B) and assigned a different community number to each module for enrichment analysis (A).**

**Table 1. Quick mapping of each analytic branch to the main figures.**

| Biological claim | Analytic branch | Primary figure(s) |
| --- | --- | --- |
| Node-level hubs / centrality | igraph / MST | Figs 3–12 |
| Variance-driven gene loadings | PCA / t-SNE / MDS | Figs 13–16 |
| Module-level co-expression | WGCNA | Figs 17–22 |
| Preprocessing / batch diagnostics | SVA / PVCA | S1–S7 Figs |

## WGCNA

**Constructing gene co-expression networks.** The specifics of network construction (TOM computation, clustering, dynamic tree cut settings, and module merging) are described in Methods ("WGCNA: construction and parameters"). Results below focus on the modules prioritized for enrichment and their biological interpretation (Figs 17–22).

Fig 17 presents the gene clustering dendrogram generated using hierarchical clustering based on the topological overlap matrix. In this dendrogram, each leaf represents an individual gene, branches indicate groups of highly interconnected genes, and distinct modules are defined by assigning colors to merged branches according to their level of co-expression.

To assess module–trait associations, we used gene significance (GS) measures. In the resulting module–trait association table (Table 1, S1 File), each row corresponds to a module eigengene, and each column represents a trait of interest. The table is color-coded to indicate the direction and strength of correlation—green for negative and red for positive associations. Each cell also reports the exact correlation coefficient and associated p-value.

**Table 2. Gene Ontology functional annotation after Fig 8.**

| Community | ID | Description | GeneRatio | BgRatio | p-value | p.adjust | q-value |
|---|---|---|---|---|---|---|---|
| 2 | GO:0006406 | mRNA export from nucleus | 3/35 | 67/20870 | 0.000192 | 0.049663 | 0.040746 |
| 3 | GO:0032608 | interferon-beta production | 5/95 | 69/20870 | 0.000016 | 0.011366 | 0.011106 |
| 26 | GO:0006119 | oxidative phosphorylation | 8/84 | 156/20870 | 0.0 | 0.00031 | 0.000278 |
| 30 | GO:0051967 | negative regulation of synaptic transmission, glutamatergic | 1/7 | 10/20870 | 0.00335 | 0.043447 | 0.017449 |
| 11 | GO:0018107 | peptidyl-threonine phosphorylation | 2/5 | 127/20870 | 0.000363 | 0.034159 | 0.010348 |
| 32 | GO:0071248 | cellular response to metal ion | 9/54 | 208/20870 | 0.0 | 0.0 | 0.0 |
| 5 | GO:0048814 | regulation of dendrite morphogenesis | 3/18 | 72/20870 | 0.000031 | 0.005462 | 0.003765 |
| 20 | GO:1903078 | positive regulation of protein localization to plasma membrane | 2/5 | 72/20870 | 0.000117 | 0.010846 | 0.00396 |
| 19 | GO:0071805 | potassium ion transmembrane transport | 2/3 | 230/20870 | 0.00036 | 0.02279 | 0.002329 |
| 42 | GO:1904659 | glucose transmembrane transport | 3/16 | 126/20870 | 0.000114 | 0.011793 | 0.008826 |
| 22 | GO:0006695 | cholesterol biosynthetic process | 2/11 | 60/20870 | 0.00044 | 0.027769 | 0.016923 |
| 13 | GO:0010959 | regulation of metal ion transport | 6/14 | 449/20870 | 0.0 | 0.00013 | 0.000071 |
| 14 | GO:0060078 | regulation of postsynaptic membrane potential | 3/13 | 149/20870 | 0.000097 | 0.010119 | 0.005427 |
| 9 | GO:0015871 | choline transport | 2/9 | 16/20870 | 0.00002 | 0.008484 | 0.004725 |
| 31 | GO:0007163 | establishment or maintenance of cell polarity | 2/3 | 228/20870 | 0.000354 | 0.023005 | 0.005216 |
| 17 | GO:1904385 | cellular response to angiotensin | 2/3 | 26/20870 | 0.000004 | 0.001207 | 0.000141 |
| – | GO:0007214 | gamma-aminobutyric acid signaling pathway | 4/56 | 33/20870 | 0.000002 | 0.002136 | 0.001908 |
| 7 | GO:0048015 | phosphatidylinositol-mediated signaling | 5/36 | 195/20870 | 0.00002 | 0.013988 | 0.010596 |
| 1 | GO:0048489 | synaptic vesicle transport | 2/6 | 45/20870 | 0.000068 | 0.004296 | 0.001844 |

Modules selected for downstream enrichment analysis were required to meet two key criteria: high GS and high module membership (MM) values. This approach enabled us to identify modules most strongly correlated with diagnosis, along with their most interconnected (hub) genes. The central hypothesis is that hub genes within disease-associated modules, which are also highly correlated with diagnosis, are likely to play biologically significant roles in the disease process. In Fig 18, the highest peaks correspond to modules showing the strongest association with diagnosis. Following the integration of clinical trait data into the weighted gene co-expression network, we identified the top 25 modules most significantly correlated with disease status, along with their corresponding p-values (Table 1, S1 File; Fig 18).

In the subsequent analysis, we focused on the *GS* and *MM* values of the top disease-associated modules. Specifically, these included: yellow ($GS = 0.70$, $p = 3.5 \times 10^{-187}$), steel blue ($0.50$, $5.5 \times 10^{-14}$), purple ($0.75$, $4.8 \times 10^{-120}$), green yellow ($0.60$, $1.3 \times 10^{-60}$), royal blue ($0.58$, $6.5 \times 10^{-35}$), orange red3 ($0.46$, $1.2 \times 10^{-06}$), turquoise ($0.49$, $4.2 \times 10^{-113}$), pink4 ($0.49$, $4.6 \times 10^{-06}$), brown4 ($0.72$, $1.6 \times 10^{-22}$), and midnight blue ($0.51$, $8 \times 10^{-32}$) (Fig 19). In Fig 20, we display a gene dendrogram highlighting the key modules selected through this analysis. Using this method, we successfully reduced the initial set of 144 modules (Table 2, S1 File) to a focused subset of 10 modules for further investigation.

Before performing Gene Ontology (GO) enrichment analysis on the final 10 modules, we briefly examined the relationships among them. To visualize these relationships, we constructed a multidimensional scaling (MDS) plot using topological overlap matrix (TOM) dissimilarity as input (Fig 21B). In this plot, each point represents a gene, color-coded according to its module assignment. The modules form distinct "finger-like" projections, with the most influential hub genes located at the fingertips.

To validate the MDS visualization, we correlated the module eigengenes with the t-SNE coordinates of each gene (Fig 21C). For instance, the brown and yellow modules—both significantly associated with diagnosis—occupy distinct regions of the t-SNE plot, consistent with the divergent directions of their corresponding fingers in the MDS plot.

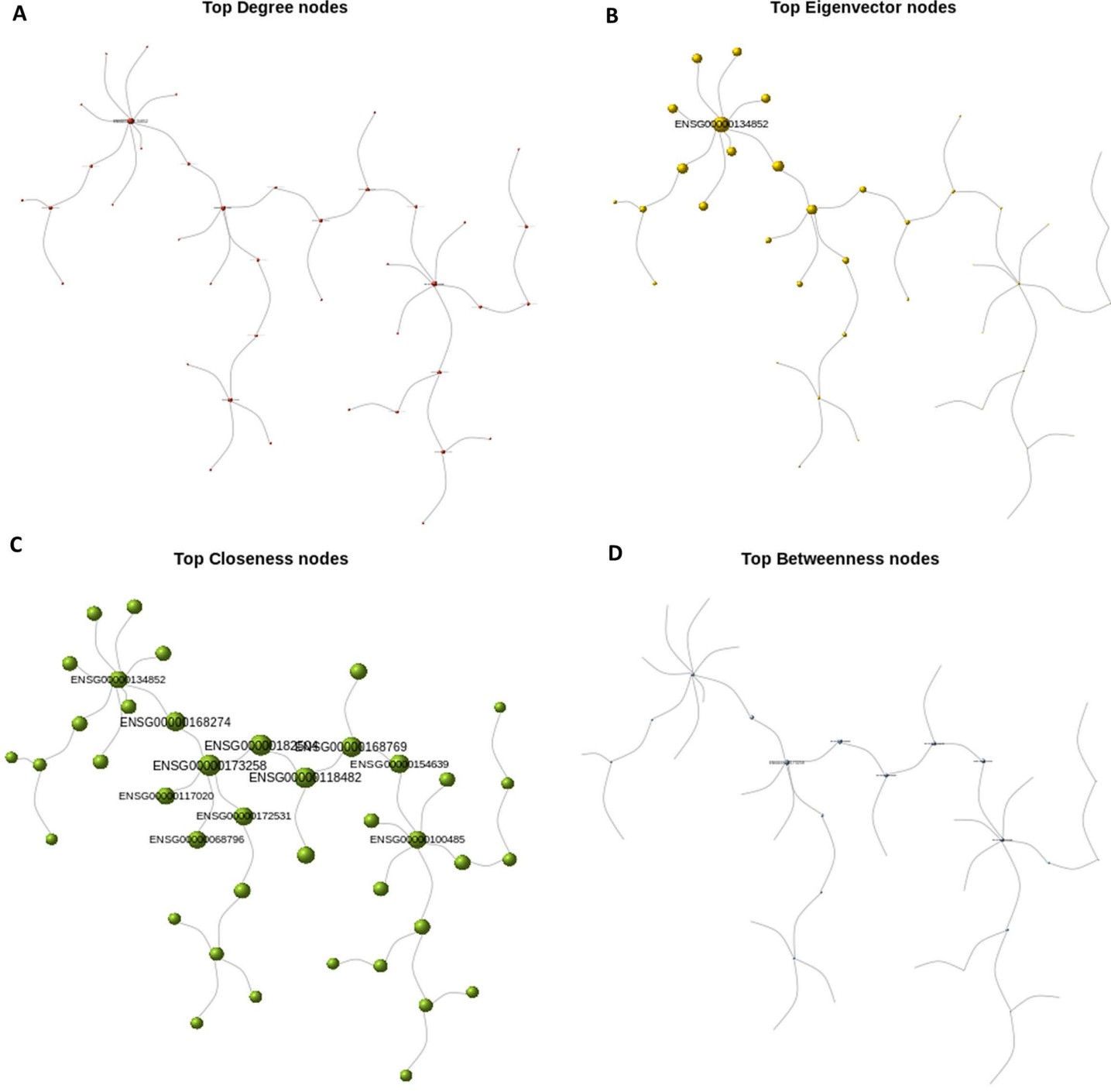

**Fig 9. Scaled centrality visualizations for Control network.** (A) top degree; (B) top eigenvector; (C) top closeness; (D) top betweenness. For each panel the node diameter is proportional to the corresponding centrality score.

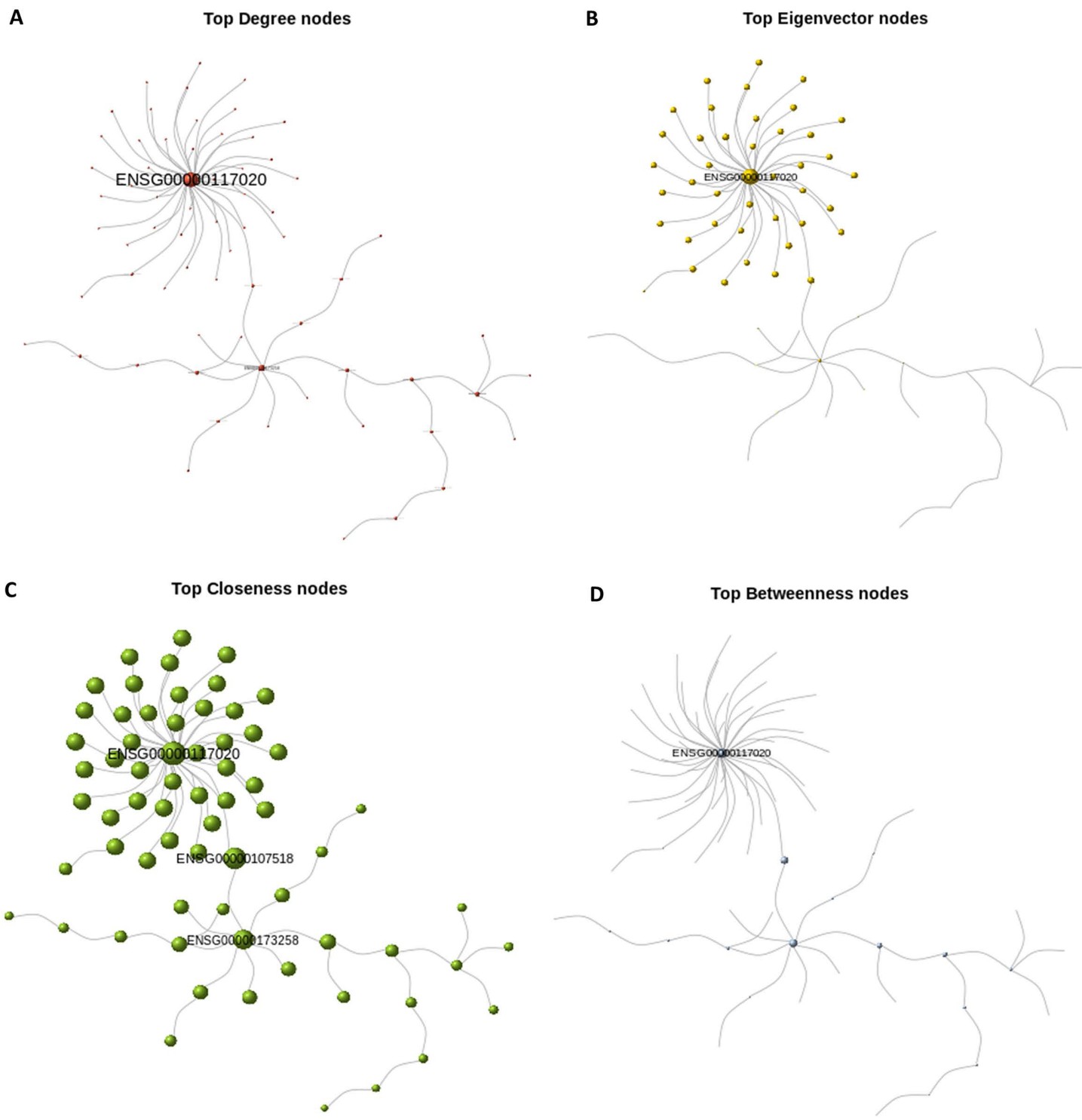

**Fig 10. Scaled centrality visualizations for Pathogenic network.** Panels (A–D) correspond to top degree, eigenvector, closeness, and betweenness centrality respectively.

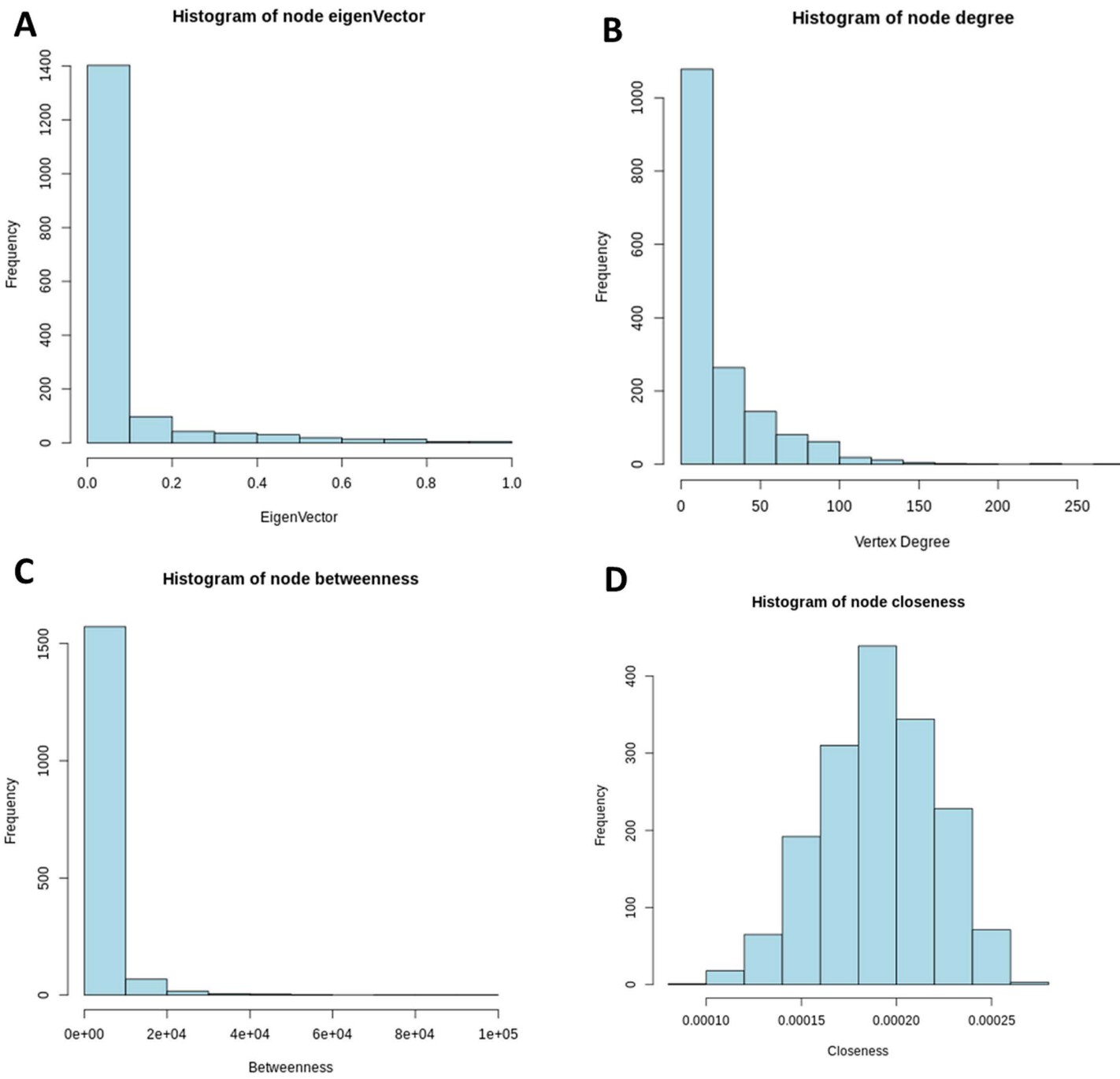

**Fig 11. Distribution histograms of centrality metrics (control).** Notably, in panel **(B)**, the degree distribution approximates a power-law behavior, characterized by the majority of nodes having fewer than 20 connections, while a small number of nodes exhibit high connectivity, with more than 150 connections. This pattern is indicative of a scale-free topology, a hallmark of many biological networks.

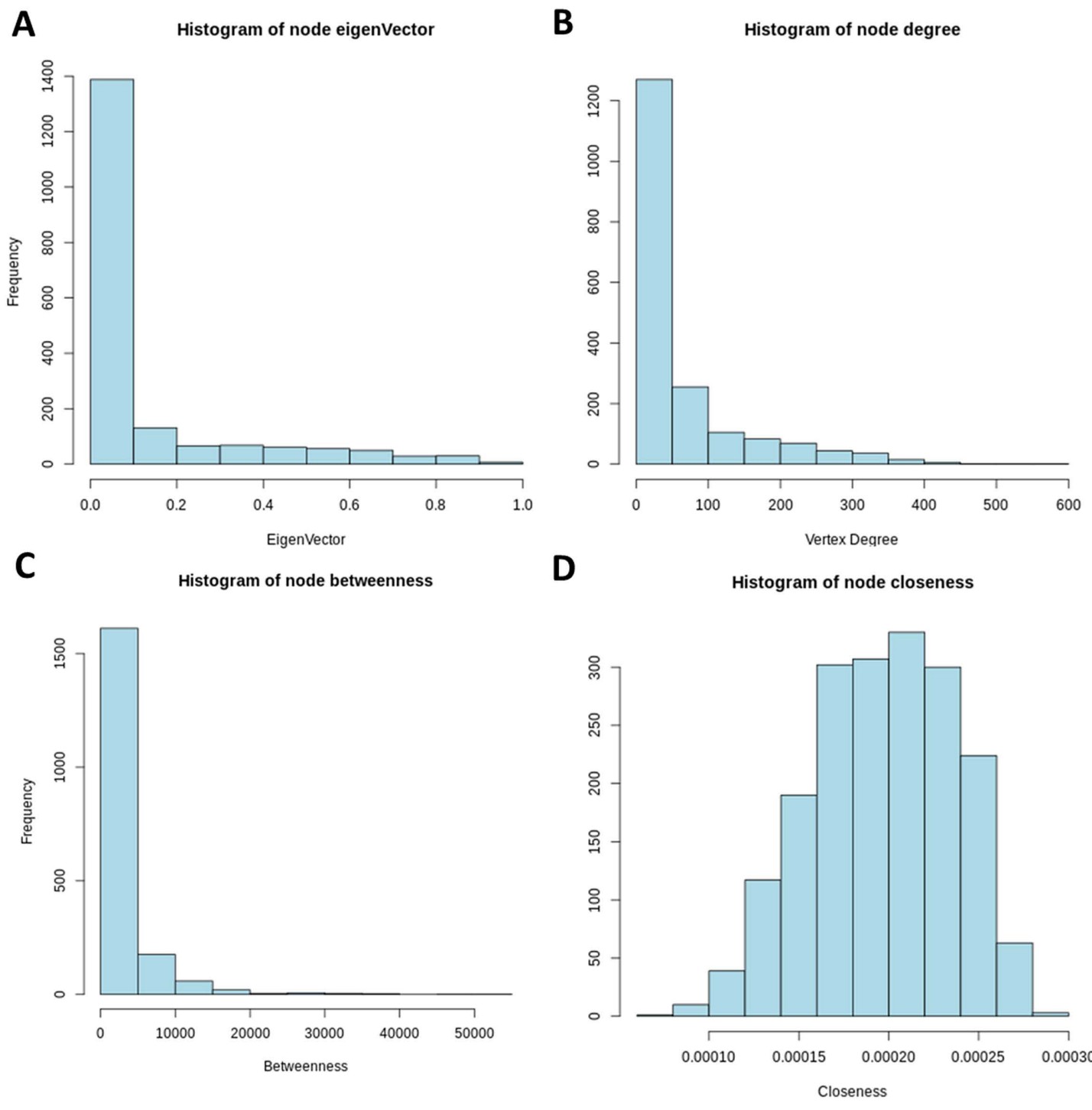

**Fig 12. Histograms of network centrality measures in schizophrenia.**

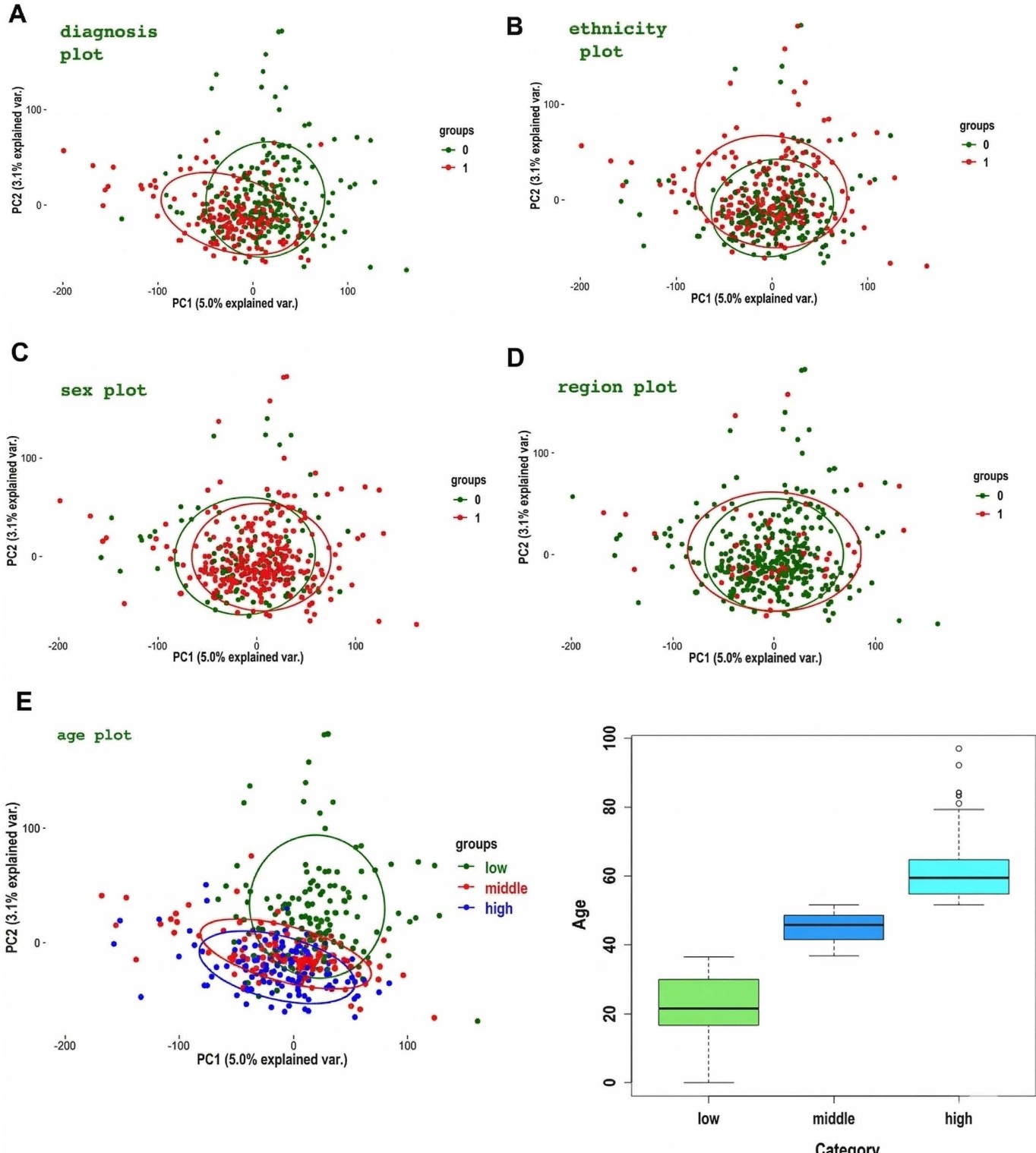

**Fig 13. PCA plots with variable factor arrows and grouping variable ellipses.**

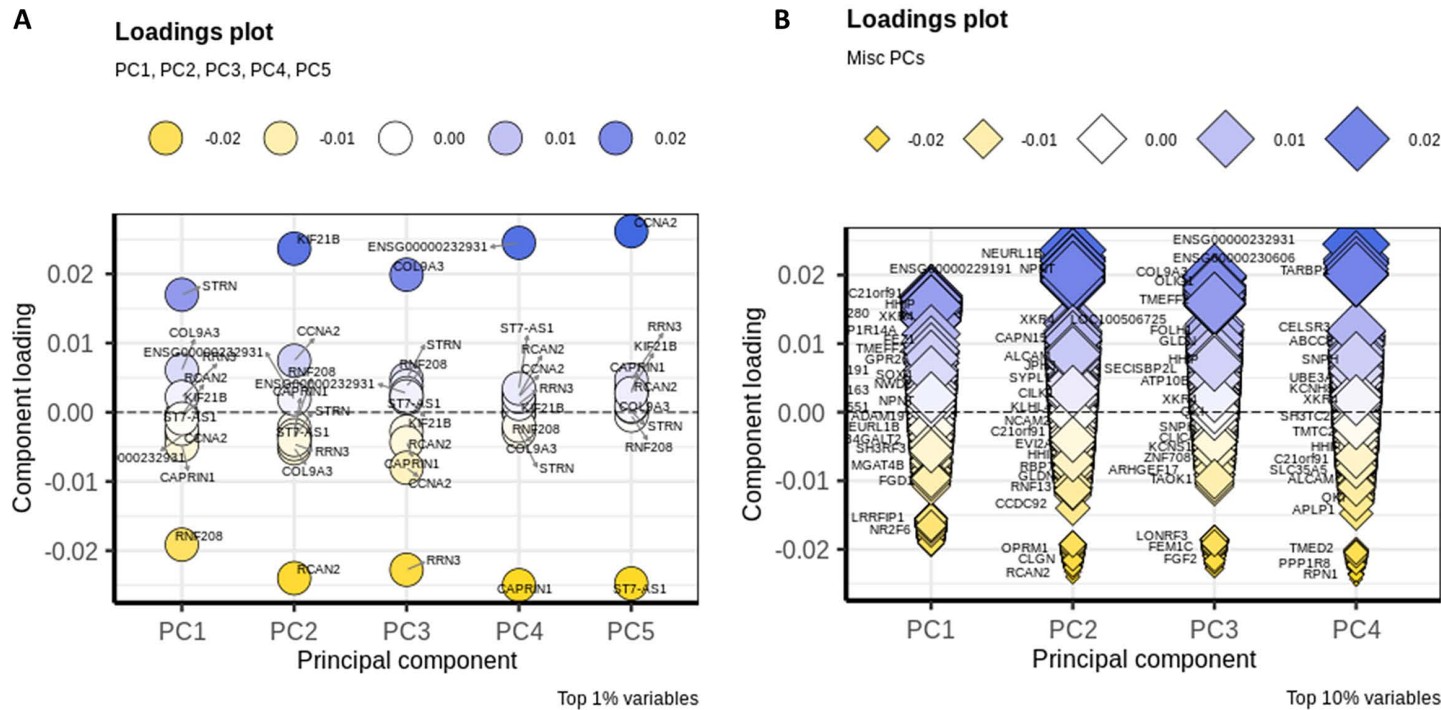

**Fig 14. Loading plots. (A)** Shows the top 1% of genes contributing to the first five principal components (PCs); (B) displays the top 10% of contributing genes.

Additionally, Fig 21A presents a correlation heatmap between individual genes and module eigengenes, further supporting the observed module structure. Our objective is to identify hub genes—those near the tips of the MDS projections—that are not only central within their own modules but also exhibit strong correlations across multiple modules, indicating potential regulatory importance.

**Gene enrichment.** Enrichment tests for WGCNA-derived modules were performed using clusterProfiler and ReactomePA as described in Methods ("Enrichment analysis") and in the consolidated WGCNA Methods subsection. Below we report the functional themes and significant pathways observed across the prioritized modules. To characterize the functional mechanisms of the targeted disease modules, selected hub genes were mapped to Bioconductor's `org.Hs.eg.db` package for the discovery of potential functional annotations. GO functional annotation revealed that the hub genes were enriched in the glutamine family amino acid catabolic process; nuclear-transcribed mRNA catabolic process, nonsense mediated decay; citrulline metabolic process, arginine catabolic process; regulation of the Wnt signaling pathway; cytosol to endoplasmic reticulum transport, antigen processing and presentation of exogenous peptide antigen via MHC class I; regulation of histone H3-K27 methylation; DNA methylation on cytosine, nuclear-transcribed mRNA poly(A) tail shortening; and histone H2B ubiquitination. Table 4 and Table 5 show that, based on KEGG pathway analysis, the hub genes were enriched in the mRNA surveillance pathway; the NF-kappa B signaling pathway; and cytokine–cytokine receptor interaction (Fig 22).

## Discussion

### Experimental, genetic, and bioinformatic evidence

Before delving into biological interpretation, we reiterate that pathway-level conclusions reported here derive from the intersection of three orthogonal analyses (node-level igraph centrality, module-level WGCNA enrichment, and PC-driven

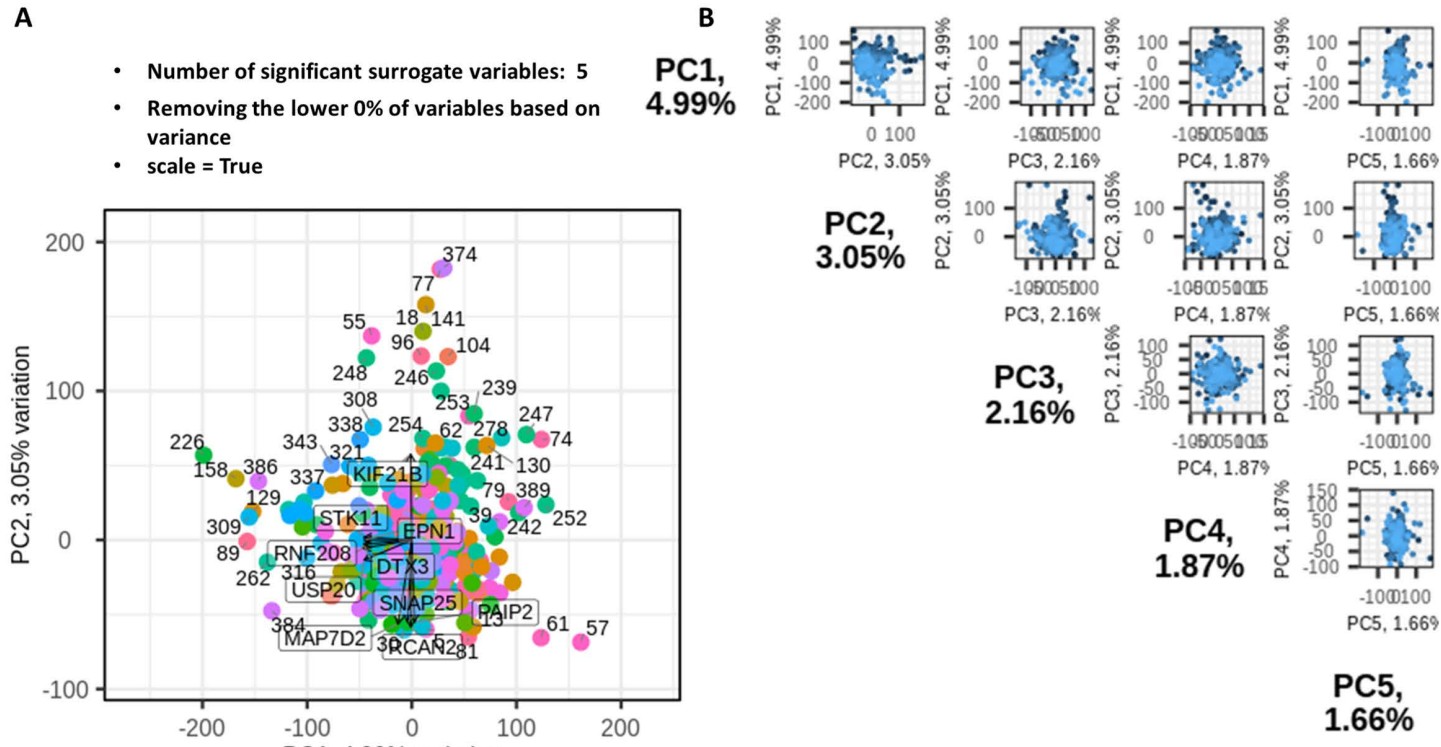

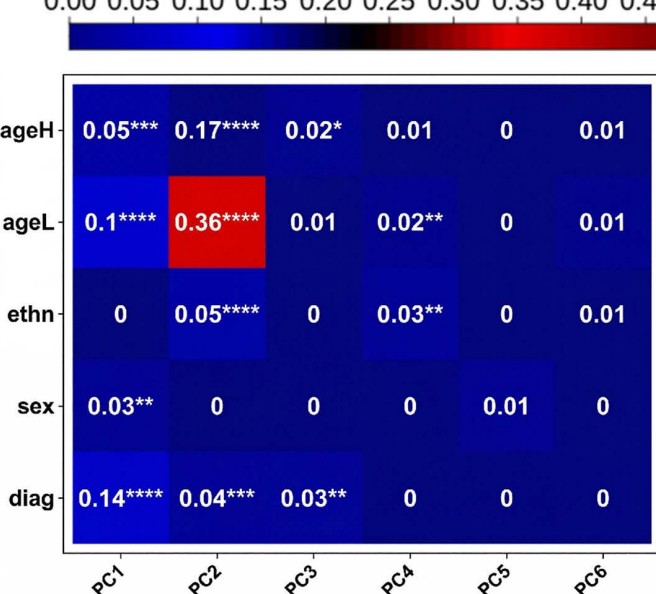

**Fig 15. Comparison of PC1 and PC2.** Both genes and samples are projected onto the same two-dimensional space. Gene contributions are represented as arrows, where the direction indicates the gene's influence on the principal components and the length reflects the magnitude of that contribution. The angles between vectors illustrate the degree of correlation among genes **(A)**. The first five principal components collectively explained approximately 14% of the total variance in the dataset **(B)**.

**Table 3. Genes with the highest positive and negative contributions to each of the top five principal components (as identified from the loading plots in Fig 14) were subjected to Reactome pathway gene set enrichment analysis.**

| PC | ID | Description | GeneRatio | BgRatio | pvalue | p.adjust | qvalue | geneID |
|---|---|---|---|---|---|---|---|---|
| 1 | R-HSA-264642 | Acetylcholine Neurotransmitter Release Cycle | 1/2 | 17/10867 | 0.003126435 | 0.01563217 | 0.01324956 | SNAP25 |
| 1 | R-HSA-181429 | Serotonin Neurotransmitter Release Cycle | 1/2 | 18/10867 | 0.003310190 | 0.01655095 | 0.01401758 | SNAP25 |
| 1 | R-HSA-181430 | Norepinephrine Neurotransmitter Release Cycle | 1/2 | 18/10867 | 0.003310190 | 0.01655095 | 0.01401758 | SNAP25 |
| 1 | R-HSA-888590 | GABA synthesis, release, reuptake and degradation | 1/2 | 19/10867 | 0.003493929 | 0.01746964 | 0.01478560 | SNAP25 |
| 1 | R-HSA-212676 | Dopamine Neurotransmitter Release Cycle | 1/2 | 23/10867 | 0.004228714 | 0.02114357 | 0.01790565 | SNAP25 |
| 1 | R-HSA-210500 | Glutamate Neurotransmitter Release Cycle | 1/2 | 24/10867 | 0.004412368 | 0.02206184 | 0.01867367 | SNAP25 |
| 2 | R-HSA-6788467 | IL-6-type cytokine receptor ligand interactions | 2/5 | 17/10867 | 0.00002297153 | 0.00011486 | 0.00009732 | IL6ST/OSMR |
| 2 | R-HSA-6783589 | Interleukin-6 family signaling | 2/5 | 24/10867 | 0.00004655860 | 0.00023279 | 0.00019727 | IL6ST/OSMR |

gene loadings); method-specific provenance for the principal claims is indicated in Methods ("Analytic responsibilities and figure mapping") and in the figure captions.

Relevant literature robustly supports the enrichment of synaptic, immune, and epigenetic processes in schizophrenia, consistent with the functional modules identified in this study. Genes such as SNAP25, highlighted in the PCA-based enrichment analysis, are essential for synaptic vesicle exocytosis and neurotransmitter release, and have been implicated in schizophrenia through transcriptomic and proteomic studies, though GWAS-level evidence is more limited [19]. Functional disruptions in glutamatergic and GABAergic signaling, as reflected in the GO terms (e.g., "negative regulation of synaptic transmission, glutamatergic" and "GABA signaling pathway"), are strongly supported by pharmacological and neurophysiological evidence linking NMDA receptor hypofunction and parvalbumin-positive interneuron deficits to the disorder's cognitive and negative symptoms [20]. Immune-related findings—such as enrichment in interferon-beta production and cellular responses to metal ions—align with studies reporting increased pro-inflammatory cytokines (e.g., IL-6, IFN-$\gamma$) and complement C4–related MHC variation as major schizophrenia risk factors [21,22]. Similarly, enrichment in mRNA export and histone H3-K27 methylation is supported by transcriptome-wide and epigenome-wide analyses that show widespread alterations in RNA metabolism, chromatin structure, and DNA methylation in schizophrenia brains [23,24].

Large-scale RNA-seq studies of postmortem schizophrenia (SCZ) brain consistently find dysregulation of RNA processing pathways. For example, Darby et al. (2016) [25] performed RNA-seq in hippocampus and orbitofrontal cortex from hundreds of SCZ and control brains and found increased expression of ribosomal/translation genes but decreased expression of neuronal and synaptic genes. Notably, gene-set analyses in that study showed strong enrichment of RNA-related processes: messenger RNA processing/export and protein synthesis pathways were among the most consistently altered across regions and disorders. Similarly, Choudhury and colleagues analyzed RNA editing (an RNA metabolism process) in multiple cohorts of postmortem cortex and report a global reduction in A-to-I RNA editing in SCZ, especially at 3'-UTRs of mitochondria-related genes [26]. These transcriptome-wide results—spanning RNA splicing, editing, and turnover—indicate widespread disruption of RNA metabolism in SCZ cortex, which naturally implicates nucleocytoplasmic mRNA export machinery; for example, GO terms such as "mRNA export from nucleus" are enriched in these dysregulated gene sets. In summary, multiple recent postmortem omics studies (RNA-seq, methylation arrays, ChIP/ATAC

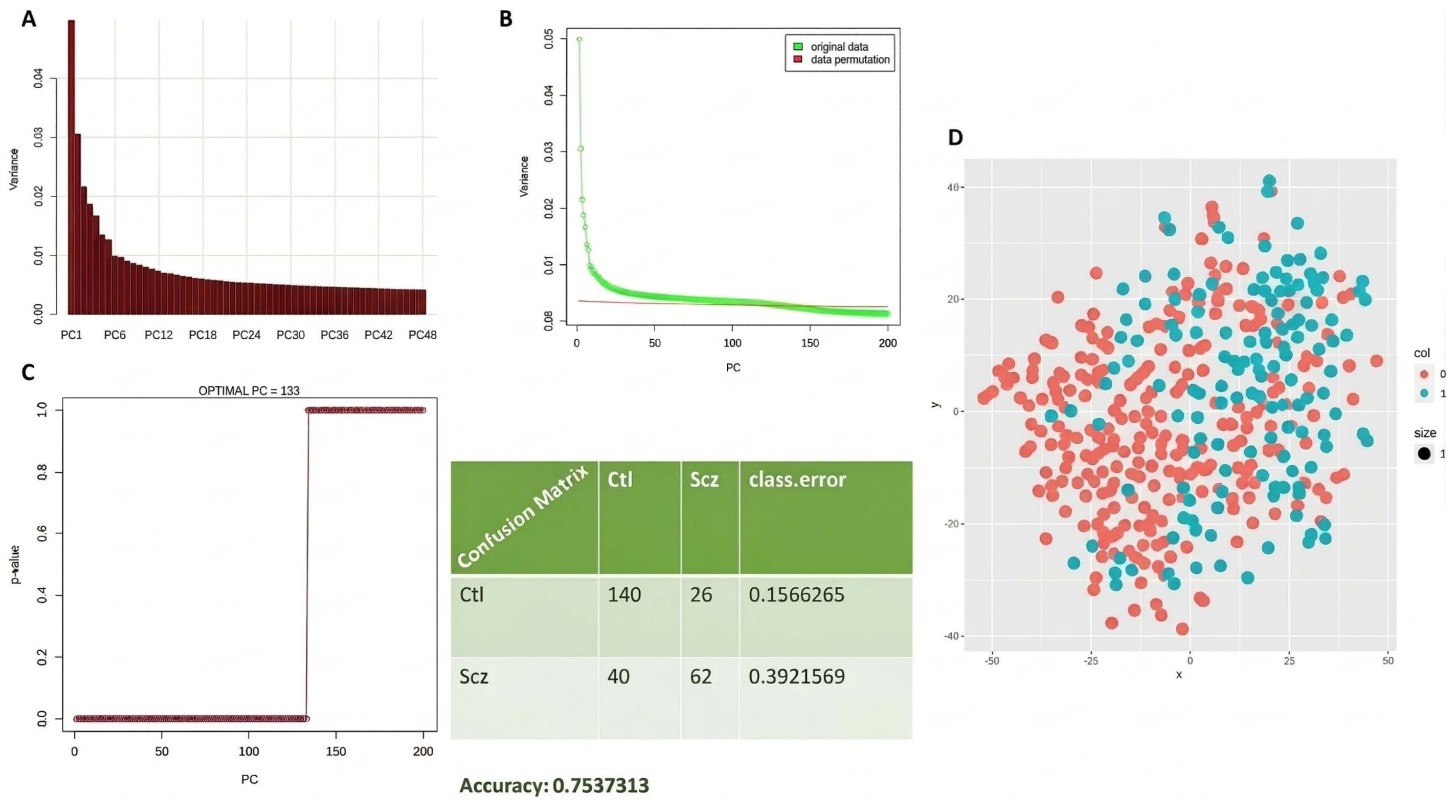

**Fig 16. PC selection and t-SNE workflow. (A)** Scree plot of observed principal component variances. **(B)** Permuted (null) variance plotted together with observed variance; the indicated intersection is the point where the observed variance curve meets the permuted/noise curve and therefore marks the cutoff number of PCs retained for downstream analysis. (C) random forest validation (ROC/AUC with retained PCs). (D) final t-SNE plot using selected PCs.

profiling) converge on widespread changes in RNA metabolism, chromatin regulation, and DNA methylation in SCZ brains – in line with enriched GO signals for mRNA export and H3K27 methylation in our data.

AKT3, identified in the igraph-based pathogenic network as a central but non-core regulator, has been shown in both computational and in vivo models to influence mTOR signaling, neurodevelopment, and epigenetic pathways, and is increasingly considered a promising therapeutic target in psychiatric and neurodevelopmental disorders [27]. Notably, AKT3 has been shown to modulate mRNA processing and chromatin regulation and is increasingly recognized as a schizophrenia-associated regulator with therapeutic potential [27], and it has recently emerged as a schizophrenia risk gene and signaling hub. For example, GWAS and pathway analyses rank the AKT3 locus among the top schizophrenia signals and show that dozens of risk-linked AKT3 SNPs map to key neural pathways [28]. Network analyses similarly identify AKT3 as a high-centrality (hub) node in schizophrenia-related gene networks [29]. In vivo studies support a causal role: Akt3-null mice exhibit prefrontal/hippocampal learning deficits and dramatically reduced cortical AKT/mTORC2 signaling (with no change in Akt1/2), indicating a nonredundant, dominant regulatory role for AKT3 in brain signaling [28]. Consistent with this, human postmortem studies report dysregulated AKT3 expression – e.g., elevated neuronal AKT3 mRNA along with insulin/AKT pathway components in schizophrenia brains, pointing to perturbation of PI3K–AKT/mTOR and related pathways [30]. Finally, AKT3 is noted as pharmacologically "druggable": it is classified as a T_chem target with many known ligands, and overlaps genes modulated by antipsychotics [31]. Together, these data implicate AKT3 in schizophrenia-specific molecular pathways and network regulation, and suggest it may be a promising therapeutic target.

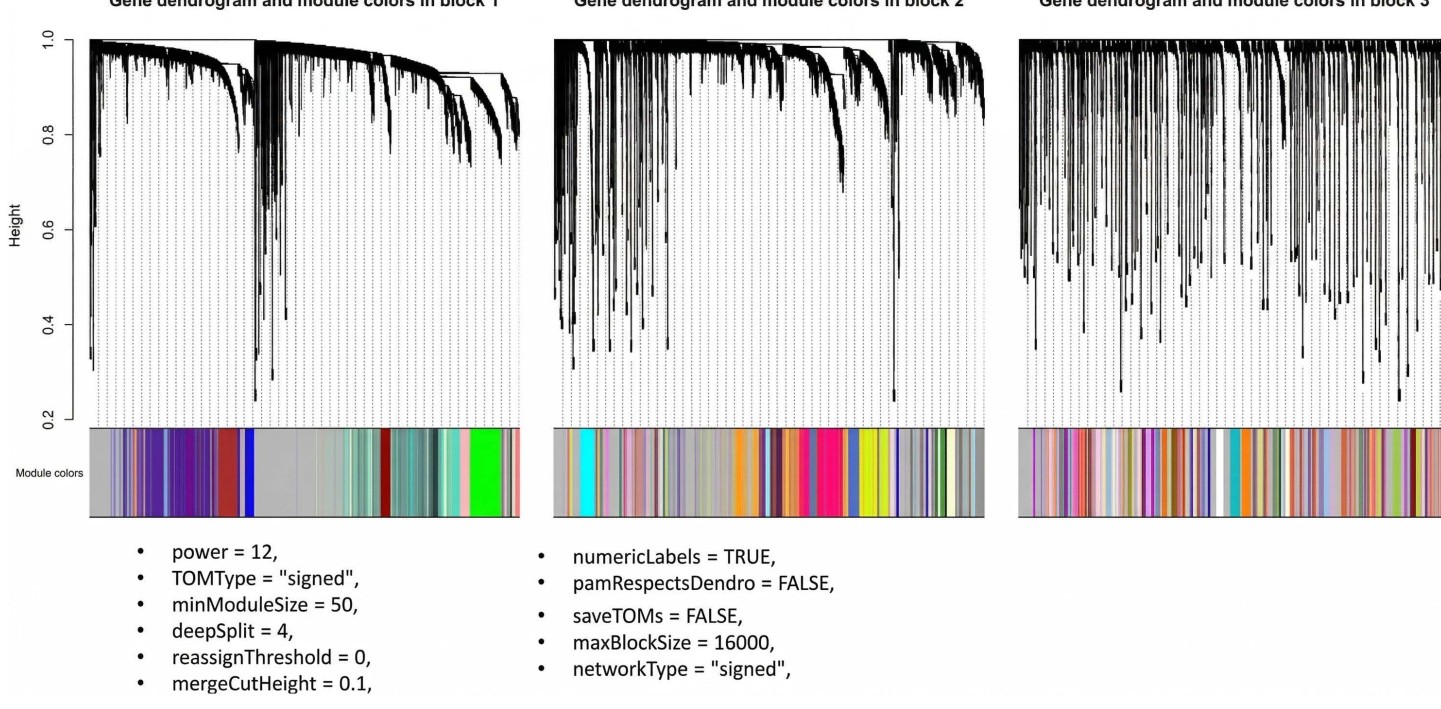

**Fig 17. Dendrogram and module colors.**

## Validation against genetic and functional-genomic resources

To assess whether our prioritized hub genes and disease-associated modules overlap established genetic risk sets, we compared our top hub genes and module gene lists to: (i) the SCHEMA exome sequencing results [32], (ii) GWAS target genes predicted and experimentally supported by MPRA/Hi-C framework [33], and (iii) large transcriptome resources such as [34] and the recent single-cell multi-cohort analysis by Ruzicka and colleagues [35].

At the gene level we did not find a direct overlap between our two highlighted hubs (AKT3 and SNAP25) and the SCHEMA top-10 exome-wide genes [32]. Likewise, AKT3 and SNAP25 are not present among the MPRA/Hi-C prioritized gene lists [33]. Importantly, however, the module- and pathway-level concordance is strong: synaptic, immune, and RNA-processing/chromatin regulatory pathways highlighted here are consistent with those reported by PsychENCODE capstone analyses and by both bulk and single-cell postmortem transcriptome studies [8,34,35].

Transcriptomic hub-genes and genetic hit lists come from different types of evidence, so lack of gene-level overlap does not contradict shared biology: rare-variant exome studies [32] identify genes carrying high-impact coding mutations, MPRA/Hi-C experiments [33] nominate regulatory target genes often in a developmental context, and bulk or single-cell expression studies [34,35] measure steady-state RNA in heterogeneous tissue. Network "hubness" reflects transcriptional centrality (high connectivity or kME) rather than being a direct readout of mutation burden or variant-to-gene mapping, and hubs can be regulatory or downstream effectors of genetically implicated nodes. Differences in tissue (adult postmortem vs developmental models), cell-type composition, mapping strategies and statistical power therefore commonly produce modest gene-level overlap while producing strong convergence at the pathway/module level (synaptic, immune, chromatin/RNA-processing), which is what we observe here — consistent with many multi-omic schizophrenia studies that report pathway-level replication despite limited one-to-one gene overlap [36,37].

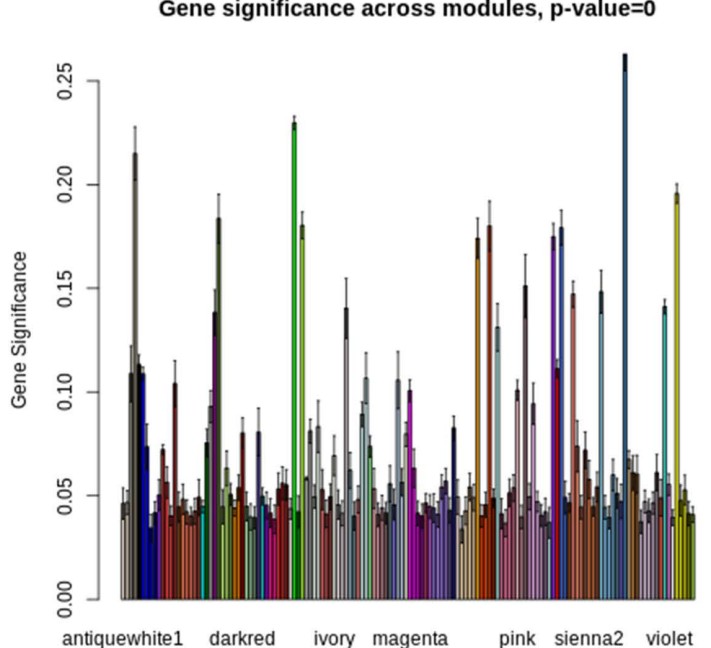

Fig 18. Bar plot of the mean gene significance across all genes in each module.

Altogether, our computational findings are well-aligned with experimental, genetic, and bioinformatic evidence across multiple domains of schizophrenia research.

## Limitations: confounding, residual variance, and generalization

Although we applied multiple preprocessing safeguards (outlier removal, PVCA-based variance decomposition, and SVA) to mitigate known and unknown sources of variation, three categories of confounding remain especially important to consider when interpreting our results: *age-related effects*, *brain-region heterogeneity*, and *technical protocol differences*. PVCA indicated that age contributed only a small proportion of explained variance (≈ 1.3%), brain region and sex were similarly modest contributors, and diagnosis accounted for ≈ 3.2% of the explained variance, whereas the vast majority of variance (≈ 92.1%) remained in the residuals. These numbers illustrate two linked points: (i) the observable, modeled covariates explain only a small slice of total variance in this large multisite collection; and (ii) a very large residual component persists after standard adjustments, and that residual variance can contain a mixture of unmodeled biology (for example, cell-type composition, medication effects, post-mortem interval, comorbidities) and technical artifacts (library preparation subtleties, sequencing center effects, RNA quality).

1. **Age.** Even when age explains a small fraction of total variance in PVCA, age-associated changes in gene expression can be non-linear and gene/pathway specific. Modules or hubs driven by genes with strong age-dependence may therefore appear associated with diagnosis if age distributions differ subtly between groups or interact with other covariates.

2. **Brain region.** PsychENCODE aggregates multiple cortical and subcortical regions; region-specific expression programs and cell-type composition differences are well known. Even modest region-associated variance can induce

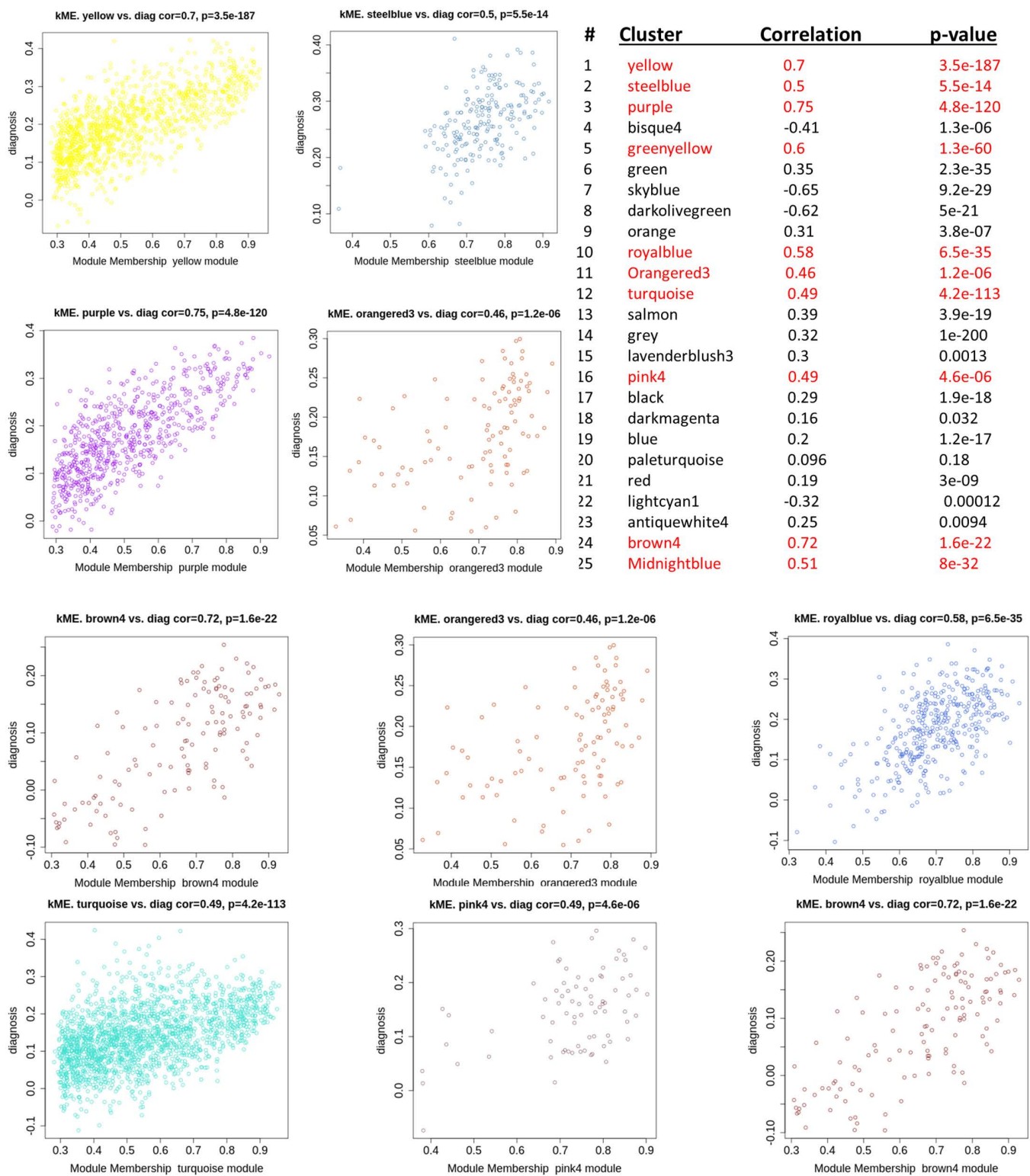

**Fig 19. Scatterplot of gene significance for diagnosis (y-axis) versus intramodular connectivity (kME) based on the module eigengene (x-axis).** Each point represents a gene within the corresponding color-coded module.

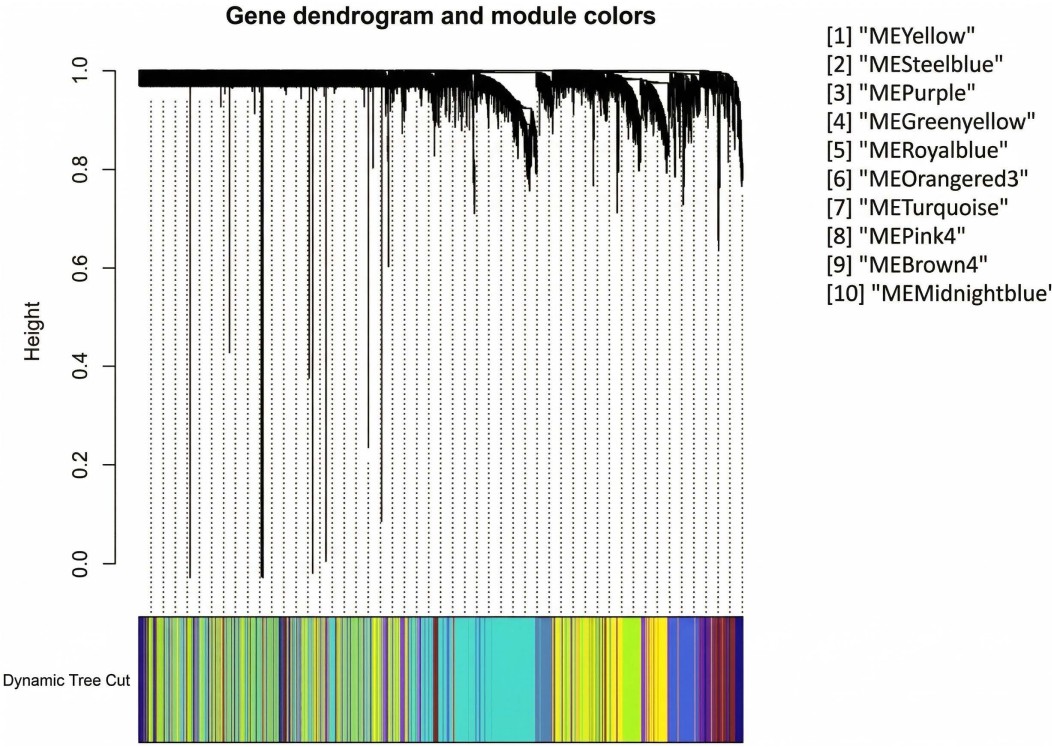

**Fig 20. Dendrogram of genes from the most relevant modules.** Branches represent clusters of highly interconnected genes, with module colors indicating the corresponding co-expression modules selected for further analysis.

false module structure when samples from different regions are analyzed together. While our PVCA suggested region explained only a small fraction of variance overall, the heterogeneity of regional sampling implies that some modules could be region-specific rather than disease-specific.

3. **Technical heterogeneity.** Our SVA checks flagged protocol heterogeneity—most notably differences in rRNA-depletion kit (Ribo-Zero Gold vs Ribo-Zero HMR)—that manifest as systematic shifts in recovered RNA species and small RNA representation. Even after removing an SV that was associated with known technical factors and passing the selection criterion that retained SVs are not strongly associated with diagnosis, residual technical effects can remain. Such residuals can concentrate in particular gene sets (e.g., small RNAs, mitochondrial transcripts, rRNA-proximal sequences) and thus bias enrichment analyses or hub detection.

A high residual variance fraction implies limited signal-to-noise for disease effect sizes and increases the risk that findings are cohort-specific. Practically, this affects different results unevenly: *pathway-level* enrichment is generally more robust to distributed residual noise than *node-level, centrality/hub* identification because enrichment aggregates signals across many genes; by contrast, single-gene centrality measures (degree, betweenness, eigenvector) are highly sensitive to small changes in the underlying correlation matrix and can be unstable when residual technical or cellular-composition effects remain. Consequently, claims about broad affected pathways (immune, synaptic, epigenetic) are more likely to generalize across independent cohorts than claims that a particular gene is the top therapeutic priority.

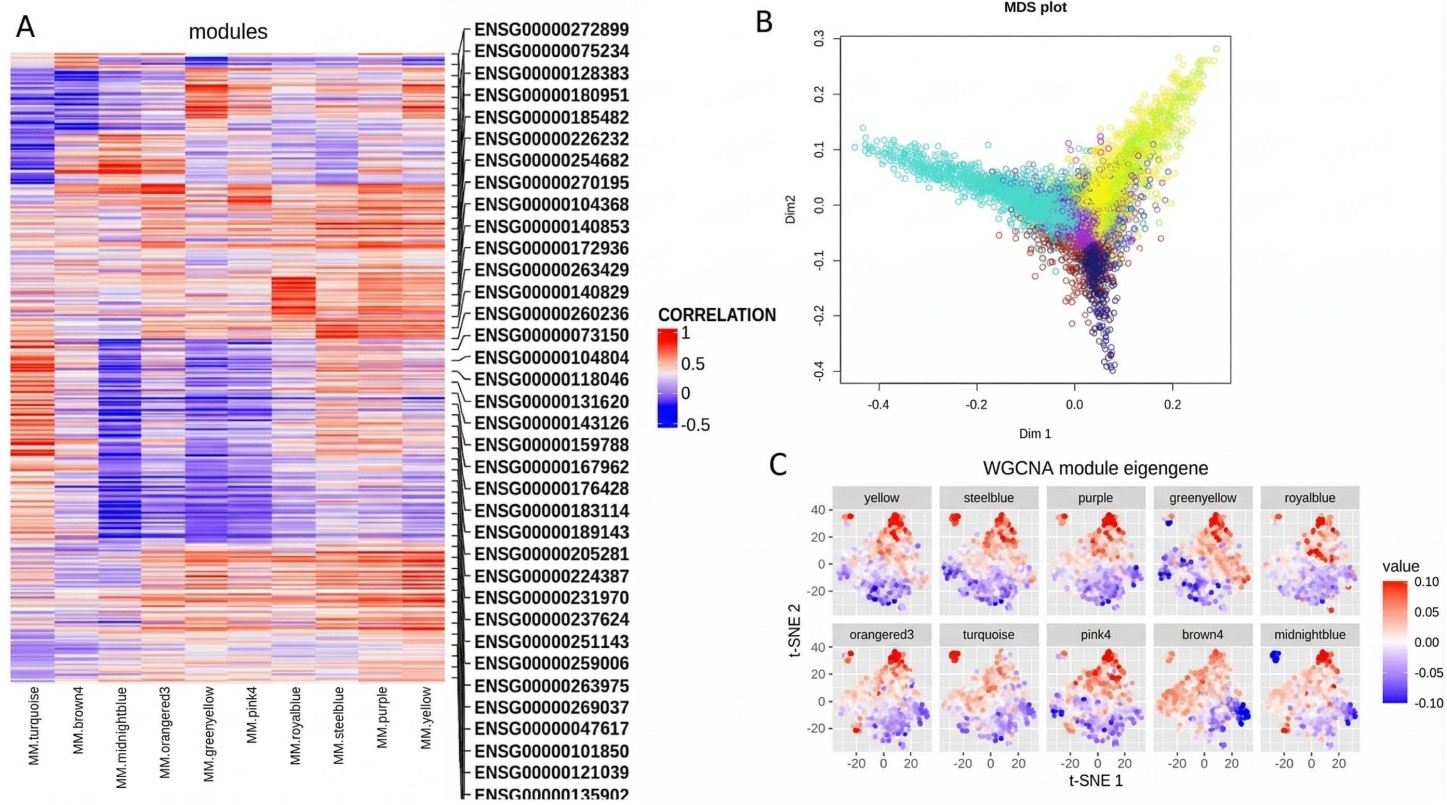

**Fig 21. Module relationships and gene-level projections. (A)** heatmap of gene – module eigengene correlations; **(B)** MDS on TOM dissimilarity with genes color-coded by module; module "fingers" highlight clusters and hub genes at their tips. **(C)** t-SNE of genes colored by module assignment (validates MDS structure).

**Table 4. GO enrichment results.**

| GO Community | ID | Description | GeneRatio | BgRatio | pvalue | p.adjust | qvalue |
|---|---|---|---|---|---|---|---|
| 1 | GO:0009065 | glutamine family amino acid catabolic process | 11/181 | 37/20973 | 0.0 | 0.0 | 0.0 |
| 2 | GO:0000184 | nuclear-transcribed mRNA catabolic process, nonsense-mediated decay | 4/116 | 48/20973 | 0.000143 | 0.00583 | 0.004947 |
| 3 | GO:0000052, GO:0006527 | citrulline metabolic process, arginine catabolic process | 7/175 | 14/20973, 17/20973 | 0.0 | 0.0 | 0.0 |
| 4 | GO:0030111 | regulation of Wnt signaling pathway | 10/94 | 350/20973 | 0.000004 | 0.001248 | 0.000964 |
| 5 | GO:0046967, GO:0042590 | cytosol to endoplasmic reticulum transport; antigen processing and presentation of exogenous peptide antigen via MHC class I | 8/63 | 18/20973, 23/20973 | 0.0 | 0.0 | 0.0 |
| 6 | GO:0061085 | regulation of histone H3-K27 methylation | 3/46 | 13/20973 | 0.000003 | 0.002116 | 0.0018 |
| 7 | GO:0032776, GO:0000289 | DNA methylation on cytosine; nuclear-transcribed mRNA poly(A) tail shortening | 7/314, 10/314 | 12/20973, 39/20973 | 0.0 | 0.0 | 0.0 |
| 9 | GO:0035323 | histone H2B ubiquitination | 1/3 | 12/20973 | 0.001716 | 0.024295 | 0.002708 |

**Table 5. KEGG pathway enrichment results.**

| KEGG Community | ID | Description | GeneRatio | BgRatio | pvalue | p.adjust | qvalue |
|---|---|---|---|---|---|---|---|
| 2 | hsa03015 | mRNA surveillance pathway | 6/44 | 85/5732 | 0.00004 | 0.006963 | 0.006866 |
| 5 | hsa04064 | NF-kappa B signaling pathway | 6/34 | 66/5732 | 0.000018 | 0.002194 | 0.001883 |
| 5 | hsa04060 | Cytokine–cytokine receptor interaction | 8/34 | 257/5732 | 0.000096 | 0.005725 | 0.004912 |

## Criteria for prioritizing therapeutic targets in schizophrenia

To identify the most promising genes for pharmacological intervention, we propose a stringent set of criteria that integrate transcriptomic, network, and phenotypic data. These criteria are designed to prioritize targets with the greatest likelihood of functional relevance, therapeutic tractability, and disease specificity:

- **Pathology-specific involvement**: The gene (or hub node) must be active in disease-associated circuits but absent or minimally involved in control networks (Figs 9 and 10). This ensures that candidate targets are not merely broadly expressed housekeeping genes but instead represent nodes with altered activity or connectivity specific to the schizophrenia condition. Such specificity reduces the risk of off-target effects and increases therapeutic precision.

- **Network centrality**: The gene must be highly correlated with multiple distinct co-expression modules (Fig 21). This indicates a central or integrative role in transcriptomic regulation, suggesting that modulating this gene could impact multiple downstream pathways simultaneously. Genes with high module connectivity are more likely to act as "bottlenecks" or master regulators of disease-relevant expression patterns.

- **Phenotypic relevance**: The gene must show a strong association with biological traits of interest (e.g., diagnosis, symptom severity, or cognitive performance; see Table 1, S1 File). Prioritizing genes that correlate with clinically meaningful endpoints ensures translational relevance and aligns therapeutic targeting with measurable patient outcomes.

- **Systems-level integration**: The gene must participate in regulatory interactions across mRNA/lncRNA–transcription factor–gene motifs, with its protein product also embedded within protein–protein interaction (PPI) networks. Such systems-level integration suggests the gene functions at the intersection of transcriptional and post-transcriptional control mechanisms. Moreover, its inclusion in PPI networks implies biochemical relevance and the feasibility of targeting it via small molecules, biologics, or protein–protein interaction modulators.

Collectively, these criteria provide a multi-dimensional framework for prioritizing targets that are not only biologically central and mechanistically relevant but also therapeutically actionable. Applying this framework to schizophrenia brain transcriptomes enables the rational selection of candidate genes with the highest potential for successful clinical translation.

To maximize translational yield from the multi-dimensional ranking described in this manuscript (network centrality, intramodular membership, phenotype correlation and druggability), we provide a concise, actionable pathway that maps evidence patterns to specific validation strategies. First, all candidates should be triaged in silico against integrated evidence and tractability resources such as Open Targets [38] and annotated for known ligands, T_chem classification, and prior human genetic support. Candidates that score highly across all axes should proceed to biochemical target-engagement assays and, where tractable chemotypes exist, testing with high-quality chemical probes or repurposing candidates following community probe standards [39]. High-confidence hubs that lack immediate tractability should instead follow an orthogonal genetic validation track using CRISPR knockout, CRISPRi, or CRISPRa, or isoform-specific oligonucleotides in disease-relevant human cellular models (for example, iPSC-derived neurons or organoids), with reproducible cellular phenotypes (electrophysiology, synaptic markers, transcriptional signatures) required prior to investing in ligand discovery [40,41]. This bifurcated strategy aligns with international probe-coverage efforts exemplified by Target2035 and with community standards for probe selection and on-target validation [42].

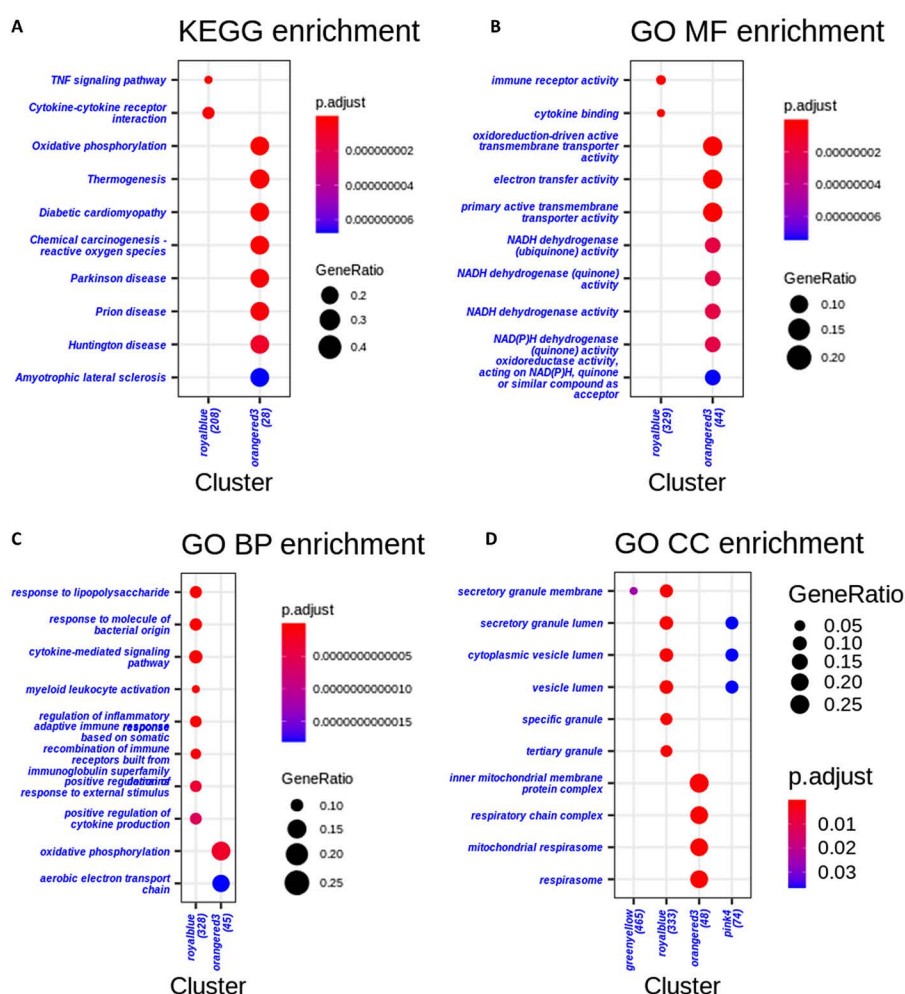

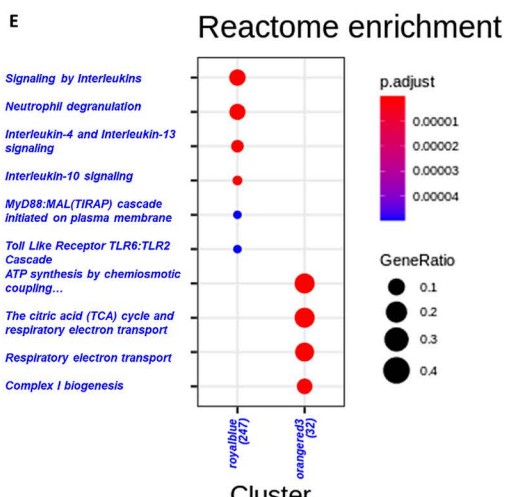

**Fig 22. Functional enrichment summary across annotation frameworks.** Panels: **(A)** KEGG pathway barplot for selected modules; **(B)** GO molecular function; **(C)** GO biological process; **(D)** GO cellular component; **(E)** Reactome pathways. Notably, the RoyalBlue module is enriched for pathways related to cytosol-to-endoplasmic reticulum transport and antigen processing and presentation of exogenous peptide antigens via MHC class **I**. The OrangeRed3 module is associated with the regulation of histone H3-K27 methylation.

Important: we do not require unanimity across axes for further follow-up; instead, we map missing evidence to appropriate next steps. For instance, a gene with strong centrality, module membership and phenotype correlation but low tractability should be advanced for genetic perturbation prior to chemistry, whereas a druggable gene without phenotype correlation should be prioritized for replication and deeper expression/isoform analysis (single-cell, independent cohorts) before experimentation. Because the manuscript explicitly compares healthy and pathogenic networks, we operationalize differential prioritization: candidates must show elevated centrality or module relevance in the pathogenic network relative to control to reduce the risk of perturbing core, healthy processes. As an example, AKT3 (ENSG00000117020), identified here as a central node in the pathogenic network, exemplifies a gene that would enter a combined track of genetic validation in iPSC models followed by targeted biochemical validation given its partial tractability and literature support.

To make these rules transparent and reproducible we recommend the following minimal decision Table 6:

Finally, we embed explicit replication gates in the pipeline: progression to resource intensive work (ligand discovery, animal studies) requires concordant module-level enrichment, independent cohort replication or orthogonal assay confirmation. These additions make the translational mapping explicit, reduce the chance of chasing cohort-specific artifacts, and align the manuscript's computational outputs with a clear, experimentally tractable validation path.

## Conclusions

### Method-specific biological signatures

Each analytical method revealed distinct biological processes, underscoring their complementary strengths in characterizing schizophrenia-associated mechanisms.

For instance, DEG analysis using the igraph framework primarily highlighted disruptions in cell transporter activity including cellular responses to metal ions, regulation of protein localization to the plasma membrane, potassium and glucose transmembrane transport, metal ion transport, and choline transport. Additionally, it identified dysfunction in synaptic transmission, such as negative regulation of glutamatergic signaling, dendrite morphogenesis, postsynaptic membrane potential, GABAergic signaling, phosphatidylinositol-mediated signaling, and synaptic vesicle transport.

In contrast, WGCNA identified significant epigenetic and immune-related modules, including regulation of histone H3-K27 methylation, histone H2B ubiquitination, cytosol-to-endoplasmic reticulum transport, antigen processing and presentation via MHC class I, NF-$\kappa$B signaling, and cytokine–cytokine receptor interactions (Fig 22). Notably, a substantial number of WGCNA-derived modules were enriched in mRNA-related processes, such as nonsense-mediated decay, poly(A) tail shortening, mRNA surveillance, and DNA methylation on cytosine. Additionally, modules associated with

**Table 6. Operational decision framework linking multi-dimensional ranking evidence to experimental validation strategy.**

| Class | Evidence pattern | Recommended next actions |
|---|---|---|
| **Rank A** | High centrality in disease and low centrality in control; high module membership (MM); strong phenotype correlation; tractable (T_chem or known ligand). | In silico triage; biochemical target engagement; chemical probe or repurposing assays; iPSC-derived neuronal phenotyping; in vivo follow-up if validated. |
| **Rank B** | High centrality, module membership, and phenotype correlation; low or uncertain tractability. | Genetic perturbation (CRISPR, CRISPRi, CRISPRa, or ASO) in iPSC models; require reproducible cellular phenotype before ligand discovery or biologics development. |
| **Rank C** | Moderate or discordant evidence; module-level enrichment without strong node-level centrality. | Orthogonal replication (independent cohorts, single-cell or isoform checks); module validation prior to experimental investment. |

altered metabolic pathways—including glutamine family amino acid catabolism, citrulline metabolism, and arginine catabolism—were also detected.

PCA-based enrichment analysis on the other hand, captured key neurotransmitter release cycles and immune regulatory pathways, including Chk1/Chk2 (Cds1)-mediated inactivation of the Cyclin B:Cdk1 complex and the activation and mitochondrial translocation of BAD proteins, linking neurotransmission and apoptotic/immune signaling.

### Future directions

In Figs 9 and 10, we visualized disease and control circuits by collapsing each network community into its node with the highest centrality. Notably, some nodes (e.g., ENSG00000173258 and ENSG00000117020) are central in both pathogenic and control networks. In such cases, targeted interventions could risk destabilizing essential regulatory processes in healthy tissues. Framed as a network control problem, the challenge is to determine the minimal subset of nodes whose perturbation can modulate a target node in one network (e.g., disease state) while leaving it unaffected in others (e.g., control state). However, this approach can be computationally intensive due to the combinatorial explosion of possible node interactions.

Previous approaches to this problem have employed various mathematical optimization techniques, including integer linear programming with branch-and-bound algorithms [43], bilevel programming [44], nonlinear programming [45], mixed-integer nonlinear programming [46], and linear approximations of nonlinear models [47]. However, an alternative and computationally efficient strategy involves Boolean (binary) modeling, which is particularly well-suited for heterogeneous networks comprising mRNAs, lncRNAs, transcription factors, and DNA elements. Boolean models require no kinetic parameters and are ideal for systems with limited quantitative data. In future work, we will further investigate the druggability of key pathogenic nodes using Boolean network modeling to simulate network dynamics and identify robust intervention points.

### Supporting information

**S1 Fig. Surrogate variable analysis (SVA) selection and design matrices used for SV estimation.** Full code and model matrices are available in the Supplementary R scripts.
(TIF)

**S2 Fig. PCA of differentially expressed genes before (left) and after (right) covariate correction.**
(TIF)

**S3 Fig. Sample-level PCA and hierarchical clustering dendrogram used for outlier detection (red cut line indicates retained cluster).**
(TIF)

**S4 Fig. SVA diagnostics: boxplots of retained SV values by diagnosis and coefficient estimates testing SV association with diagnosis.**
(TIF)

**S5 Fig. PVCA variance decomposition summary showing proportions explained by covariates, SVs, and residuals.**
(TIF)

**S6 Fig. Protocol heterogeneity diagnostics: PCA colored by rRNA depletion kit and post-adjustment PCA.**
(TIF)

**S7 Fig. Soft-threshold selection (pickSoftThreshold output) and permutation-based PC scree diagnostics used to select WGCNA power and PCs retained for t-SNE.**
(TIF)

**S1 Code. Analytical pipelines and custom scripts.** This archive contains four distinct Jupyter Notebooks used for the analysis: `SVA_Diagnostics_and_PCA_Loadings.ipynb`, `PCA_Variance_Driven_Reactome_Enrichment.ipynb`, `WGCNA_Module_Functional_Enrichment.ipynb`, and `igraph_MST_Topological_Centrality.ipynb`. (ZIP)

**S1 File. Contains extended module-trait association tables and Reactome pathway figures.**
(PDF)

**S2 File. Contains Supplementary Figures S1–S7, which detail the PVCA variance decomposition, SVA selection diagnostics, protocol-heterogeneity assessments, and pickSoftThreshold outputs.**
(PDF)

## Acknowledgments

We gratefully acknowledge the members of the PsychENCODE Consortium for their patient and thorough responses to our inquiries.

## Author contributions

**Conceptualization:** Costas Bampos.

**Data curation:** Costas Bampos.

**Formal analysis:** Costas Bampos.

**Methodology:** Costas Bampos.

**Software:** Costas Bampos.

**Supervision:** Vasileios Megalooikonomou.

**Visualization:** Costas Bampos.

**Writing – original draft:** Costas Bampos.

**Writing – review & editing:** Vasileios Megalooikonomou.

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
