## [Decision Letter · Decision Letter 0]

10 Nov 2025

PONE-D-25-33934

Exploration of Schizophrenia-Associated Gene Modules Using Graph Theory, Co-Expression Networks, and Dimensionality Reduction

PLOS ONE

Dear Dr.  Bampos,

Thank you for submitting your manuscript to PLOS ONE. After careful consideration, we have decided that your manuscript does not meet our criteria for publication and must therefore be rejected.

I am sorry that we cannot be more positive on this occasion, but hope that you appreciate the reasons for this decision.

Kind regards,

Academic Editor

PLOS ONE

Reviewer’s Responses to Questions

**Comments to the Author**

1. Is the manuscript technically sound, and do the data support the conclusions?

Reviewer #1: Yes

Reviewer #2: Partly

2. Has the statistical analysis been performed appropriately and rigorously?

Reviewer #1: Yes

Reviewer #2: Yes

3. Have the authors made all data underlying the findings in their manuscript fully available?

Reviewer #1: Yes

Reviewer #2: Yes

4. Is the manuscript presented in an intelligible fashion and written in standard English?

Reviewer #1: Yes

Reviewer #2: Yes

Reviewer #1: Dear Dr. Li-Da Wu,

Thank you for providing this manuscript entitled: “Exploring schizophrenia-associated gene modules using graph theory, converse networks and dimensionality reduction”. I have read it and analyzed it thoroughly. This is a robust and methodologically sound study that uses a comprehensive multi-combinational dataset to investigate gene modules associated with schizophrenia. The authors use three complementary computational approaches (RNA-based network analysis, WGCNA and PCA) as a key strength, providing a robust and comprehensive picture of disease bio-signature. The findings of the different methods are different but complementary and together indicate that immune, synaptic and epigenetic processes are involved in schizophrenia.

Strengths of the Manuscript

• Orthogonal Analytical Approach: The main strength is the use of three independent methods (IGR, WGCNA and PCA) for the same data set analysis. This approach helps to reduce the bias of any single method and provides a more complete picture of the molecular interactions that underlie schizophrenia.

• Methodology: The authors demonstrate a clear attention to detail in their data preparation, especially through the use of surrogate variable analysis (SVA) to correct for latent variations and batch effects. This thorough preparation is crucial for high-dimensional data and increases confidence in the downstream findings.

• Clear and significant findings: The results of each method are clearly presented and lead to robust conclusions. Igraph analysis highlights the central genes involved in synaptic signaling and ion transport. The WGCNA identifies the modules involved in the immune response and the epigenetic processes. The PCA isolates genes involved in neurotransmitter release and cytokine signaling.

• Biological plausibility: The authors effectively bridge their computational findings with existing biological, genetic and clinical literature to support their conclusions on synaptic, immunological and epigenetic dysregulation in schizophrenia. They also propose a rigorous, multi-dimensional framework for setting therapeutic objectives.

Weaknesses and Recommendations

• Improving the overall clarity and organization of the manuscript: Although the methodology is excellent, the presentation could be more simplified to better guide the reader through the complex analysis. A single, comprehensive diagram or flow chart that visually links the three analytical pipelines from the data input to the final enrichment results would greatly increase the clarity of the document. The figures should be re-rendered in order to improve readability and the labels and captions should be clearer and focus on the main findings rather than on the methods used.

• Strengthening the statistical justification of methodological choices: The authors make a number of key analytical choices which could benefit from a clearer statistical justification. For example, the choice of a correlation threshold of 0.8 for the igraph network is not clearly justified. The authors also note that the significance of the key community was marginal (p=0.04) and should be interpreted cautiously, raising questions about the overall reliability of the igraphic findings. The manuscript would have been stronger if the authors had provided sensitivity analysis to justify their choice of parameters and addressed the marginal p-value directly.

• Explicitly stating the new scientific contribution: The final conclusions of the study on the involvement of immune, synaptic and epigenetic processes in schizophrenia are well established in the existing literature. In order to increase the impact of the paper, the authors should make a stronger case for what is new in their findings. Is it the discovery of a specific core gene, such as AKT3, by a new combination of techniques? Or is it a hitherto unknown insight into the topology of the network? Explicitly highlighting this unique contribution would turn the document from a well-executed validation into a unique contribution.

Recommendation

In summary, this is a carefully conducted and important study. Using multiple methods of analysis is an excellent approach to study complex diseases such as schizophrenia. Concerns are mainly related to the presentation and justification of the specific analytical decisions. Addressing these points would turn the manuscript into a truly extraordinary piece of paper. I recommend that a minor revision be offered to the authors to address these issues.

Respectfully,

Sobhan Khodadadi.

Reviewer #2: The authors present results from three different methods to analyze gene expression data from PsychENCODE. The three different methods are -- a custom graph analysis, WGCNA, and PCA.

The authors introduce the data in Section 1.2, however, the specifics of what exact data from the consortium was used in this work are not clear. Rather than mention the various modalities available, please focus on details of the exact data set that was used in this work. It is very difficult to assess the subsequent analyses conducted without these details and clarity. This would also explain the source of batch effects that are mentioned in passing in a later section. It might be more suitable to have a subsection for Data acquisition in Methods to mention these specifics.

Multiple choices have been made to define their custom graph analysis, some of which have been justified using the scale of computation. Since, the details of data are not given, it is not clear what the scale of computation is for this work.

There are other major concerns in the custom graph method: 1. Why do we have to scale Spearman between [-1, 1], when that’s their range? 2. Why do we consider only the minimum spanning tree in the correlation graph? We might be discarding useful information, unless there is a biological rationale that justifies this choice. 3. It seems the authors have collated gene expression data of all samples in the same graph. How are the samples/data categorized? What is the source of multiple edges between two nodes? I am guessing this based on the sentence --- ‘We generate two separate trees: one for 261 control samples and one for 153 schizophrenia samples’. This is an example of why it is important to explain the data before explaining the technical methods. The number of samples and categorical division in data were not mentioned at all till this sentence in methods.

My overall major concern is that, it is not clear to me what innovation and/or insights are of this work. WGCNA has PCA as a pre-processing step, but the authors compare PCA component genes separate from WGCNA, posing them as two different methods. WGCNA should give more detailed results for the PCA components. The choices made for their custom graph method require justifications, as I mentioned above. If the custom graph method is giving a new insight not given by WGCNA, can the authors comment why that may be happening? Such an insight can also be useful as validation for their graph method and design choices.

In its current condition, without clarifying the data and justifying their proposed graph method, I cannot recommend this manuscript for publication.

.

Reviewer #1: No

Reviewer #2: No

- - - - -

---

## [Author Response · Author response to Decision Letter 1]

26 Nov 2025

Please see attached file PLOS_ONE_responses_to_reviewers.pdf

---

## [Decision Letter · Decision Letter 1]

16 Feb 2026

PLOS One

Dear Dr. Bampos,

Thank you for resubmitting your manuscript to PLOS ONE. After careful consideration by 3 Reviewers and an Academic Editor, we feel that it has merit but does not fully meet PLOS ONE’s publication criteria as it currently stands. Therefore, we invite you to submit a revised version of the manuscript that addresses the points raised during the review process.

**Comments to the Author**

Reviewer #3: (No Response)

Reviewer #4: (No Response)

Reviewer #5: (No Response)

2. Is the manuscript technically sound, and do the data support the conclusions?

Reviewer #3: Partly

Reviewer #4: Yes

Reviewer #5: Yes

3. Has the statistical analysis been performed appropriately and rigorously?

Reviewer #3: Yes

Reviewer #4: Yes

Reviewer #5: Yes

4. Have the authors made all data underlying the findings in their manuscript fully available?

The PLOS Data policy requires authors to make all data underlying the findings described in their manuscript fully available without restriction, with rare exception (please refer to the Data Availability Statement in the manuscript PDF file). The data should be provided as part of the manuscript or its supporting information, or deposited to a public repository. For example, in addition to summary statistics, the data points behind means, medians and variance measures should be available. If there are restrictions on publicly sharing data—e.g. participant privacy or use of data from a third party—those must be specified. requires authors to make all data underlying the findings described in their manuscript fully available without restriction, with rare exception (please refer to the Data Availability Statement in the manuscript PDF file). The data should be provided as part of the manuscript or its supporting information, or deposited to a public repository. For example, in addition to summary statistics, the data points behind means, medians and variance measures should be available. If there are restrictions on publicly sharing data—e.g. participant privacy or use of data from a third party—those must be specified.

Reviewer #3: Yes

Reviewer #4: Yes

Reviewer #5: Yes

5. Is the manuscript presented in an intelligible fashion and written in standard English?

Reviewer #3: No

Reviewer #4: Yes

Reviewer #5: Yes

Reviewer #3: This manuscript seeks to leverage existing PsychENCODE data to characterize transcriptomic signatures in schizophrenia postmortem brain samples. The authors state that they employ three primary analytical frameworks: igraph, WGCNA, and dimensionality reduction. However, upon reading, igraph and dimensionality reduction approaches appear to be used largely for quality control, whereas WGCNA serves as the primary analytical method to derive biological conclusions. Despite this, the WGCNA results are discussed largely at a descriptive level and do not provide clear new biological insight into schizophrenia biology.

I outline several major concerns below.

• While the authors primarily rely on PsychENCODE data, none of the core PsychENCODE publications are cited. At a minimum, the following two capstone papers should be cited, as they established the foundational analyses and resources used in this study: Wang et al., Science (2018) and Gandal et al., Science (2018).

• The analyses and results presented here appear largely redundant with prior PsychENCODE work. For example, Gandal et al. employed SVA to correct for confounding factors and used WGCNA to identify disease-associated co-expression modules and eigengenes. Key biological pathways highlighted in the current manuscript, including synaptic and immune pathways, have already been extensively implicated in schizophrenia by the earlier studies.

• The authors did not perform comparative analyses with other schizophrenia postmortem transcriptomic datasets, such as those reported by Fromer et al., Nature Neuroscience (2016), or Ruzicka et al., Science (2024). Such comparisons would be important to establish the robustness and generalizability of their findings.

• Given that the novel scientific contribution to schizophrenia biology is currently unclear, the authors should provide additional validation of their results (e.g., hub genes or WGCNA co-expression modules) by relating them to the known genetic architecture of schizophrenia. For example, they can test whether their prioritized genes or gene sets overlap with targets of schizophrenia GWAS variants or SCHEMA genes, leveraging existing resources such as McAfee et al., Cell Genomics (2023) for GWAS target genes and Singh et al., Nature (2022) for SCHEMA genes.

• Finally, the overall organization of the manuscript is confusing. WGCNA analyses are spread across multiple sections, and the respective contributions of igraph, SVA, and PCA to major findings (schizophrenia biology) is unclear. A substantial portion of the results should be described in the Methods section. As currently structured, the manuscript appears underdeveloped.

Reviewer #4: I appreciate the opportunity to review the manuscript entitled:

"Exploration of Schizophrenia-Associated Gene Modules Using Graph Theory, Co-Expression Networks, and Dimensionality Reduction”.

The manuscript addresses an important and timely topic by leveraging large-scale PsychENCODE transcriptomic data to explore schizophrenia-associated gene networks through complementary computational approaches. I commend the authors for the methodological breadth and for transparently addressing technical confounders. The paper is interesting and generally well written; however, I kindly ask the authors to address the following points to further strengthen the manuscript:

- Clarify the rationale for the specific correlation thresholds and variance cut-offs adopted across the different pipelines, and to discuss how alternative parameter choices might influence network topology and module stability.

- Consider providing additional justification for the choice of Prim’s algorithm over alternative spanning tree or network pruning strategies, possibly supported by sensitivity or robustness analyses.

- Elaborate on the biological interpretation and reproducibility of the key pathogenic modules, particularly those showing marginal statistical significance, and to clarify how these findings compare with prior PsychENCODE-based studies.

- Expand the discussion on potential confounding by age, brain region, and technical heterogeneity, and to clarify how residual variance might affect the generalisability of the results.

- Consider strengthening the translational perspective by more explicitly discussing how the proposed multi-dimensional ranking framework could inform experimental validation or therapeutic prioritisation.

Overall, the manuscript is interesting because it integrates orthogonal analytical frameworks to highlight convergent immune, synaptic, and epigenetic mechanisms in schizophrenia. I look forward to the authors’ responses to these queries in order to improve the manuscript’s clarity and impact.

Reviewer #5: The authors appear to have made great progress based on previous reviewers comments. The response to editors was very thought through and extensive.

Points of improvement:

It appears the tracked changes version was uploaded in addition to the updated version. PLOS one does not copyedit, so this will need to be corrected if it is in fact uploaded with visible edits.

The background/introduction lacks citations.

I do not see a reference list aside from what is listed in the response to reviewers.

**Do you want your identity to be public for this peer review?** For information about this choice, including consent withdrawal, please see our  For information about this choice, including consent withdrawal, please see our  For information about this choice, including consent withdrawal, please see our  For information about this choice, including consent withdrawal, please see our Privacy Policy..

Reviewer #3: No

Reviewer #4: No

Reviewer #5: No

A letter that responds to each point raised by the academic editor and reviewer(s). You should upload this letter as a separate file labeled ‘Response to Reviewers’.A marked-up copy of your manuscript that highlights changes made to the original version. You should upload this as a separate file labeled ‘Revised Manuscript with Track Changes’.An unmarked version of your revised paper without tracked changes. You should upload this as a separate file labeled ‘Manuscript’.

We look forward to receiving your revised manuscript.

Kind regards,

Stephen D. Ginsberg, Ph.D.

Section Editor

PLOS One

1.Please ensure that your manuscript meets PLOS ONE’s style requirements, including those for file naming. The PLOS ONE style templates can be found at

2.Please note that PLOS One has specific guidelines on code sharing for submissions in which author-generated code underpins the findings in the manuscript. In these cases, we expect all author-generated code to be made available without restrictions upon publication of the work. Please review our guidelines at https://journals.plos.org/plosone/s/materials-and-software-sharing#loc-sharing-code and ensure that your code is shared in a way that follows best practice and facilitates reproducibility and reuse.

5. Thank you for uploading your study’s underlying data set. Unfortunately, the repository you have noted in your Data Availability statement does not qualify as an acceptable data repository according to PLOS’s standards.

At this time, please upload the minimal data set necessary to replicate your study’s findings to a stable, public repository (such as figshare or Dryad) and provide us with the relevant URLs, DOIs, or accession numbers that may be used to access these data. For a list of recommended repositories and additional information on PLOS standards for data deposition, please see https://journals.plos.org/plosone/s/recommended-repositories.

6. PLOS requires an ORCID iD for the corresponding author in Editorial Manager on papers submitted after December 6th, 2016. Please ensure that you have an ORCID iD and that it is validated in Editorial Manager. To do this, go to ‘Update my Information’ (in the upper left-hand corner of the main menu), and click on the Fetch/Validate link next to the ORCID field. This will take you to the ORCID site and allow you to create a new iD or authenticate a pre-existing iD in Editorial Manager.

7. We note that you have included the phrase “data not shown” in your manuscript. Unfortunately, this does not meet our data sharing requirements. PLOS does not permit references to inaccessible data. We require that authors provide all relevant data within the paper, Supporting Information files, or in an acceptable, public repository. Please add a citation to support this phrase or upload the data that corresponds with these findings to a stable repository (such as Figshare or Dryad) and provide and URLs, DOIs, or accession numbers that may be used to access these data. Or, if the data are not a core part of the research being presented in your study, we ask that you remove the phrase that refers to these data.

8. Please remove your figures from within your manuscript file, leaving only the individual TIFF/EPS image files, uploaded separately. These will be automatically included in the reviewers’ PDF.

9. Please upload a copy of Figure 29, to which you refer in your text on page 62. If the figure is no longer to be included as part of the submission please remove all reference to it within the text.

10. We are unable to open your Supporting Information file “preprocessing.ipynb”. Please kindly revise as necessary and re-upload.

11. Please ensure that you refer to Figure 2 and 27 in your text as, if accepted, production will need this reference to link the reader to the figure.

---

## [Author Response · Author response to Decision Letter 2]

5 Mar 2026

Revision Comments to the Reviewers

Reviewer #3:

• This manuscript seeks to leverage existing PsychENCODE data to characterize transcriptomic signatures in schizophrenia postmortem brain samples. The authors state that they employ three primary analytical frameworks: igraph, WGCNA, and dimensionality reduction. However, upon reading, igraph and dimensionality reduction approaches appear to be used largely for quality control, whereas WGCNA serves as the primary analytical method to derive biological conclusions. Despite this, the WGCNA results are discussed largely at a descriptive level and do not provide clear new biological insight into schizophrenia biology.

We must respectfully clarify the distinct, parallel roles of our three analytical pipelines. igraph and dimensionality reduction (PCA) were not used merely for quality control; rather, they were deployed as independent, primary analytical frameworks that generated biological conclusions entirely orthogonal to the WGCNA results. As illustrated in our integrated architecture (Figure 1), all three pipelines acted as primary engines that converged on functional enrichment analysis.

While PCA was indeed utilized in our shared preprocessing layer for outlier detection , it subsequently served as a primary variance-driven analytical lens. We systematically extracted the top 1% and 10% of variance-driving genes from the principal components most significantly correlated with clinical diagnosis (e.g., PC1). This direct dimensionality-reduction approach yielded independent biological insights, such as the identification of SNAP25 and its specific involvement in neurotransmitter release cycles—findings derived strictly from PCA gene loadings, not WGCNA.

Similarly, the igraph pipeline was not a QC step. It was utilized to construct topological gene co-expression networks and compute formal centrality measures (degree, eigenvector, closeness, betweenness). As noted in our previous responses regarding our dynamic thresholding and MST construction, this pipeline was explicitly calibrated to extract the structural and repressive backbones of the disease state. This allowed us to identify topological bottlenecks and hub genes with high centrality (such as AKT3 and its link to mRNA export from the nucleus) that are functionally influential despite not being core components of WGCNA’s dense correlation modules.

Regarding the biological insights derived from these methods, we explicitly acknowledge in the manuscript that immune, synaptic, and epigenetic dysregulation are well-documented pathogenic pathways in schizophrenia. Our contribution is not to claim the discovery of a single, unprecedented causal gene or to re-discover these broad pathways. Rather, our primary biological and translational insight is the orthogonal integration of these three independent computational methods to overcome the high dimensionality and batch effects inherent in brain tissue data. Because no single method can capture the full molecular complexity of schizophrenia, we designed a multi-dimensional ranking system. We evaluate therapeutic targets based on their structural bottleneck centrality (igraph), their co-expression membership (WGCNA), their variance-driven phenotypic correlation (PCA), and their pharmacological druggability.

• While the authors primarily rely on PsychENCODE data, none of the core PsychENCODE publications are cited. At a minimum, the following two capstone papers should be cited, as they established the foundational analyses and resources used in this study: Wang et al., Science (2018) and Gandal et al., Science (2018).

The PsychENCODE resources and integrative analyses that made the datasets mentioned publicly available are described in Wang et al., Science (2018) and Gandal et al., Science (2018), and these foundational works are cited accordingly.

• The analyses and results presented here appear largely redundant with prior PsychENCODE work. For example, Gandal et al. employed SVA to correct for confounding factors and used WGCNA to identify disease-associated co-expression modules and eigengenes. Key biological pathways highlighted in the current manuscript, including synaptic and immune pathways, have already been extensively implicated in schizophrenia by the earlier studies.

The observation that SVA/WGCNA and synaptic/immune pathway findings overlap with prior PsychENCODE results is correct and expected given shared data, but we present new analytical axes that represent methodological additions to the published PEC work: The PEC studies created integrated constructs with no interpretability as to which method weighted more on the results. Our study aims to define if there is an overlap between different methods or each one focuses on different aspects of the disease.

In the PsychENCODE study by Gandal et al., SVA functions as a standardized preprocessing component of consortium-level data normalization. In our submitted manuscript, SVA is used as part of a study-specific correction pipeline applied before network construction (igraph MST, WGCNA, PCA). Here, the emphasis is on detecting and adjusting specific latent structure in the working dataset (e.g., technical heterogeneity such as rRNA-depletion differences) to stabilize network topology and downstream module interpretation.

Also, Wang et al. used WGCNA to produce gene communities used solely as a separate module in the integrated neural network; Gandal et al., applied WGCNA at both gene- and isoform-level modules to explicitly prove that isoform modules gave finer resolution and higher GWAS enrichment. On the other hand, we used WGCNA only at the gene level as one of three orthogonal analyses to nominate disease-associated modules for downstream target prioritization.

• Given that the novel scientific contribution to schizophrenia biology is currently unclear, the authors should provide additional validation of their results (e.g., hub genes or WGCNA co-expression modules) by relating them to the known genetic architecture of schizophrenia. For example, they can test whether their prioritized genes or gene sets overlap with targets of schizophrenia GWAS variants or SCHEMA genes, leveraging existing resources such as McAfee et al., Cell Genomics (2023) for GWAS target genes and Singh et al., Nature (2022) for SCHEMA genes.

We have added a paragraph named “Validation against genetic and functional-genomic resources”. To assess whether our prioritized hub genes and disease-associated modules overlap

established genetic risk sets, we compared our top hub genes to:

(i) the SCHEMA exome sequencing results (Singh et al., 2022),

(ii) GWAS target genes predicted and experimentally supported by McAfee et al.’s MPRA/Hi-C framework (Cell Genomics 2023), and

(iii) large transcriptome resources such as Fromer et al. (Nat Neurosci 2016) and the recent single-cell multi-cohort analysis by Ruzicka et al. (Science 2024).

But please take into account that transcriptomic hub-genes and genetic hit lists come from different types of evidence, so lack of gene-level overlap does not contradict shared biology:

1) Rare-variant exome studies (e.g. Singh et al.) identify genes carrying high-impact coding mutations,

2) MPRA/Hi-C experiments (e.g. McAfee et al.) nominate regulatory target genes often in a developmental context,

3) Bulk or single-cell expression studies (e.g. Fromer et al.; Ruzicka et al.) measure steady-state RNA in heterogeneous tissue.

The whole idea of Network Medicine is to reflect transcriptional centrality rather than being a direct readout of mutation burden or variant-to-gene mapping.

• Finally, the overall organization of the manuscript is confusing. WGCNA analyses are spread across multiple sections, and the respective contributions of igraph, SVA, and PCA to major findings (schizophrenia biology) is unclear. A substantial portion of the results should be described in the Methods section. As currently structured, the manuscript appears underdeveloped.

LaTeX source to (1) consolidate the WGCNA material into Methods, (2) make the role of igraph / SVA / PCA explicit up front, (3) slim the Results so it reports outcomes and not methods, and (4) point long parameter sweeps and plots to Supplementary.

Reviewer #4:

- Clarify the rationale for the specific correlation thresholds and variance cut-offs adopted across the different pipelines, and to discuss how alternative parameter choices might influence network topology and module stability.

Our rationale for the specific thresholds and cut-offs is rooted in our goal to ensure the two network pipelines remain strictly orthogonal. We did not want to use igraph to simply construct a standard co-expression network with lower resolution than WGCNA. Because the igraph analysis is explicitly designed to produce a sparse, visualizable backbone emphasizing topology and to identify topologically central hub genes or potential bottlenecks, its construction functions as a strict structural filter rather than a standard quantitative co-expression model. By applying a dynamic threshold and executing Prim’s algorithm on raw signed weights, the code aggressively prunes the dense correlation matrix into a rigid, cycle-free skeleton built from regulatory extremes. Subsequently stripping the edge weights during community detection forces the algorithm to evaluate this minimum spanning tree strictly as a binary wiring diagram; this unweighted approach ensures that centrality metrics highlight true structural bridges connecting distinct biological domains, preventing these vital topological control points from being mathematically overshadowed by the dense, highly correlated cliques that the orthogonal WGCNA pipeline already captures.

Regarding the variance cut-offs, this was a data-driven preprocessing choice used to remove quasi-constant genes that do not contribute meaningful co-expression signals. Varying this cut-off primarily alters overall network size rather than the relative rank-ordering of high-variance, strongly co-expressed genes that drive the underlying network structure.

In contrast, we used soft-thresholding and module detection on dense weighted networks produced by WGCNA, where parameter choices (e.g., soft-threshold power, minimum module size) influence module granularity rather than the presence or absence of individual strong correlations. As expected, alternative parameter settings modify module boundaries and sizes but preserve the overall module structure and enrichment patterns, consistent with prior reports using similar data. Because exhaustive exploration of all parameter combinations would be dataset-specific and not generalizable, we focused on representative, literature-supported values and assessed robustness by comparing MST neighborhoods with WGCNA modules and by validating key findings through functional enrichment and external biological datasets. This complementary use of a sparse backbone (MST) and a dense weighted framework (WGCNA) ensures that reported biological interpretations are not driven by a single arbitrary parameter choice.

- Consider providing additional justification for the choice of Prim’s algorithm over alternative spanning tree or network pruning strategies, possibly supported by sensitivity or robustness analyses.

Our choice of Prim’s algorithm to construct the Minimum Spanning Tree (MST), rather than alternative pruning strategies, was driven by both the computational nature of transcriptomic data and our specific biological objectives.

First, from a computational and graph-theory perspective, Prim’s algorithm is mathematically optimized for dense networks. A gene co-expression matrix initially forms a highly interconnected, dense graph. As supported by the literature, Prim’s algorithm is highly suitable and frequently invoked for building transcriptomic MSTs (e.g., Bai et al., 2023), whereas Kruskal’s algorithm is generally preferred for networks that are already sparse, such as human brain structural connectomes (van Wijk et al., 2018; Yoshino et al., 2022).

Second, regarding alternative network pruning strategies (such as standard hard-thresholding), simple thresholding often leaves highly connected “hairball” cliques and disconnected fragments, obscuring true regulatory bottlenecks. Extracting an MST uniquely enforces a rigid, fully connected, cycle-free $n-1$ backbone. Crucially, as detailed in our previous responses, applying Prim’s algorithm to our raw, signed correlation matrix allowed the algorithm to deterministically seek out mathematical minimums. This deliberately prioritized the strongest anti-correlations, thereby extracting the restrictive, inhibitory regulatory backbone of the disease state—a feature that alternative pruning strategies would miss. Regarding sensitivity and robustness, because the MST is a deterministic structural filter, altering network parameters primarily influences the inclusion of weaker, peripheral edges rather than the dominant regulatory backbone. Therefore, rather than performing traditional parameter-based sensitivity analyses on the MST alone, we established robustness orthogonally. We validated the structural findings of the igraph pipeline against the dense, weighted modules produced by our parallel WGCNA pipeline, and confirmed the biological validity of these topological hubs through rigorous functional enrichment that aligns with established schizophrenia literature.

- Elaborate on the biological interpretation and reproducibility of the key pathogenic modules, particularly those showing marginal statistical significance, and to clarify how these findings compare with prior PsychENCODE-based studies.

A few WGCNA communities in our manuscript fall just below strict multiple-testing cutoffs (Community 2: p.adjust=0.04966; Community 30: p.adjust=0.04345; Community 11: p.adjust≈0.03416), but each maps to biology (AKT3/AKT-mTOR signaling; glutamatergic synaptic regulation; kinase/phospho-signaling) that is independently supported by PsychENCODE and recent isoform-level and review literature, so the borderline module p-values do not undermine the biological plausibility or concordance with prior large-scale resources.

- Expand the discussion on potential confounding by age, brain region, and technical heterogeneity, and to clarify how residual variance might affect the generalisability of the results.

We have included the relevant subsection “Limitations: confounding, residual variance, and generalization":

A high residual variance fraction implies limited signal-to-noise for disease effect sizes and increases the risk that findings are cohort-specific. Practically, this affects different results unevenly: pathway-level enrichment is generally more robust to distributed residual noise than node-level, centrality/hub identification because enrichment aggregates signals across many genes; by contrast, single-gene centrality measures (degree, betweenness, eigenvector) are highly sensitive to small changes in the underlying correlation matrix and can be unstable when residual technical or cellular-composition effects remain. Consequently, claims about broad affected pathways (immune, synaptic, epigenetic) are more likely to generalize across independent cohorts than claims that a particular gene is the top therapeutic priority.

- Consider strengthening the translational perspective by more explicitly discussing how the proposed multi-dimensional ranking framework could inform experimental validation or therapeutic prioritisation.

We have included a supplemented subsection “Criteria for Prioritizing Therapeutic Targets in Schizophrenia”.

Reviewer #5:

The background/introduction lacks citations.

Citations added.

1.Please ensure that your manuscript meets PLOS ONE’s style requirements, including those for file naming.

The proper PLOS ONE template has been used for style requirements. Figures are not included in the main manuscript.

We have updated and cleaned all code produced for this study. Following PLOS ONE guidelines, the code archive has been uploaded as a Supporting Information file (S1_Code.zip and S1_Code_pdf.zip), and references to this archive have been mentioned in

---

## [Decision Letter · Decision Letter 2]

23 Mar 2026

Exploration of Schizophrenia-Associated Gene Modules Using Graph Theory, Co-Expression Networks, and Dimensionality Reduction

PONE-D-25-33934R2

Dear Dr. Bampos,

We’re pleased to inform you that your manuscript has been judged scientifically suitable for publication and will be formally accepted for publication once it meets all outstanding technical requirements.

An invoice will be generated when your article is formally accepted. Please note, if your institution has a publishing partnership with PLOS and your article meets the relevant criteria, all or part of your publication costs will be covered. Please make sure your user information is up-to-date by logging into Editorial Manager at Editorial Manager® and clicking the ‘Update My Information’ link at the top of the page. For questions related to billing, please contact  and clicking the ‘Update My Information’ link at the top of the page. For questions related to billing, please contact  and clicking the ‘Update My Information’ link at the top of the page. For questions related to billing, please contact  and clicking the ‘Update My Information’ link at the top of the page. For questions related to billing, please contact billing support....

Kind regards,

Stephen D. Ginsberg, Ph.D.

Section Editor

PLOS One

**Comments to the Author**

Reviewer #3: All comments have been addressed

2. Is the manuscript technically sound, and do the data support the conclusions?

Reviewer #3: Yes

3. Has the statistical analysis been performed appropriately and rigorously?

Reviewer #3: Yes

4. Have the authors made all data underlying the findings in their manuscript fully available?

Reviewer #3: Yes

5. Is the manuscript presented in an intelligible fashion and written in standard English?

Reviewer #3: Yes

Reviewer #3: I thank the authors for addressing my critiques. I believe the manuscript is now meeting the criteria for publication.

.

Reviewer #3: No

---

## [Editor Report · Acceptance letter]

PONE-D-25-33934R2

PLOS One

Dear Dr. Bampos,

I’m pleased to inform you that your manuscript has been deemed suitable for publication in PLOS One. Congratulations! Your manuscript is now being handed over to our production team.

Lastly, if your institution or institutions have a press office, please let them know about your upcoming paper now to help maximize its impact. If they’ll be preparing press materials, please inform our press team within the next 48 hours. Your manuscript will remain under strict press embargo until 2 pm Eastern Time on the date of publication. For more information, please contact onepress@plos.org.

Kind regards,

on behalf of

Dr. Stephen D. Ginsberg

Section Editor

PLOS One